

1            **Applying dynamical systems techniques to real ocean drifters**

2         Irina I. Rypina[1], Timothy Getscher[2], Lawrence Pratt[1], and Tamay Ozgokmen[3]

3         [1]Woods Hole Oceanographic Institution, 266 Woods Hole rd., Woods Hole, MA 02540

4         [2]US Navy

5         [3]Rosenstiel School of Marine and Atmospheric Science, U. of Miami, 4600 Rickenbacker

6         Causeway, Miami, FL 33149

7         Corresponding author: Irina I. Rypina, irypina@whoi.edu





**Abstract**
This paper presents the first comprehensive comparison of several different dynamical-systems-
based measures of stirring and Lagrangian coherence, computed from real ocean drifters. Seven
commonly used methods (finite-time Lyapunov exponent, trajectory path length, trajectory
correlation dimension, trajectory encounter volume, Lagrangian-averaged vorticity deviation,
dilation, and spectral clustering) were applied to 135 surface drifters in the Gulf of Mexico in
order to map out the dominant Lagrangian coherent structures. Among the detected structures
were regions of hyperbolic nature resembling stable manifolds from classical examples,
divergent and convergent zones, and groups of drifters that moved more coherently and stayed
closer together than the rest of the drifters. Many methods highlighted the same structures, but
there were differences too. Overall, 5 out of 7 methods provided useful information about the
geometry of transport within the domain spanned by the drifters.




**Significance statement**
This paper applies different Lagrangian methods to identify Lagrangian Coherent Structures
based on trajectories of 135 ocean surface drifters from the massive drifter release in the Gulf of
Mexico. To our knowledge, this is the first comprehensive comparison of the performance of
several different dynamical systems techniques with applications to real ocean drifters.















## 1. Introduction

Techniques from the dynamical systems theory can be used to study transport and exchange processes in oceanic flows (Haller, 2015; Samelson and Wiggins, 2006; Balasuriya et al., 2019; Hadjighasem et al., 2017; Filippi et al., 2021; Rypina et al., 2010 and others). In general, they aim to identify the key regions of the flow with qualitatively different Lagrangian behavior and/or to identify boundaries between them. The term Lagrangian Coherent Structures or LCS (Haller and Juan, 2000) has been adopted to refer to both such regions themselves and to their boundaries. Because different methods use different definitions of "different" and "similar," they generally yield different LCS (Balasuriya et al. 2019; Rypina et al. 2011; 2018; Hadjighasem et al., 2017).

Being Lagrangian in nature, most LCS detection methods start with the release of a set of particles or drifters within the domain of interest, and then observation of their trajectories as the particles are advected by the flow. Obtaining such trajectory datasets is straightforward in applications where the velocity fields are known from either models or observations, and this is exactly the settings in which the dynamical system approach has been used in the past. However, applying the same techniques to real ocean drifters has been a challenge simply because the drifters are rarely released in a manner that adequately spans the domain of interest.

On April 21$^{st}$ 2018, 135 near-surface CARTHE drifters were released nearly simultaneously in a roughly 10 km by 11 km domain in the northern Gulf of Mexico as part of the Submesoscale Processes and Lagrangian Analysis on the Shelf (SPLASH) experiment (Laxague et al., 2018; Solodoch et al., 2020; Lund et al., 2020). The release pattern was a nearly regular, rectangular, 11 x 12 grid with roughly 1 km average spacing between neighboring drifters. The release was





done using 7 boats and took less than 2 hours. The drifters then transmitted their positions every
5 min during the subsequent 5 days. The deployment positions and the resulting drifter
trajectories are shown in Fig. 1. Such aggressive release strategy is not typical for oceanographic
applications due to high coasts of vessels and manpower. However, it allowed populating the
domain with drifters in a manner most suitable for the dynamical systems applications. Thus, this
dataset provided a unique and long-awaited opportunity to try applying the dynamical systems
techniques to real, rather than simulated, ocean drifters and to identify the real, rather than
simulated, ocean LCS.
In this paper, seven commonly used dynamical systems techniques were applied to the real
drifter dataset from the SPLASH experiment: FTLEs, trajectory path length, trajectory
correlation dimension, encounter volume, Lagrangian-averaged vorticity deviation, dilation, and
spectral clustering. The resulting real ocean LCS were mapped and described and, when
possible, parallels were made between these observed structures and their more classical
counterparts from text-book analytic or numeric examples. The seven techniques were also inter-
compared to each other and the similarities/differences were discussed. Our choice of the seven
techniques is by no means all-inclusive and was inspired by Hadjighasem et all. (2017) who
compared a similar selection of the dynamical systems methods (plus a few more and minus the
encounter volume method) in the context of analytical and numerically-generated flows.
**2. Methods**
We start with a brief review of the 7 dynamical systems techniques that we will use.
**a) FTLEs**





One of the most commonly used LCS detection techniques is based upon FTLEs (Haller and
Yuan, 2000; Shadden et al., 2005). FTLE is the largest exponential separation rate between a
trajectory and its closest neighbors in any direction. Maximizing ridges of FTLE fields can be
used as proxies for stable (or unstable for backward-time trajectories) manifolds of hyperbolic
trajectories in time-varying fluid flows (with the additional requirement that the fastest
separation occurs in the direction normal to the ridge and is caused by the hyperbolic straining
rather than shear). Regions with small FTLEs are indicative of slow separation rates between
neighboring trajectories and often correspond to eddy cores, although for eddies with a non-
solid-body rotating regime, azimuthal shear produces spiraling structures within the core region
instead of the uniformly low FTLEs. Maps of FTLEs are very visual, and the computation of
FTLEs is straightforward, computationally inexpensive, and robust with respect to noise, which
makes FTLEs one of the most popular methods in oceanographic studies of transport and
mixing. Importantly, FTLEs are also frame-independent and thus give consistent results in any
translating or rotating reference frame (Haller 2005; 2015).
For flows where the velocity field is known from either models or observations, FTLEs ($\lambda$) can
be estimated by releasing dense regularly-spaced orthogonal grids of simulated trajectories
(Haller, 2002; Lekien and Ross, 2010; Rypina et al., 2021). This method uses 4 (in 2D) closest
neighbors to construct the Cauchy-Green tensor $G = \left(\frac{\Delta x_i}{\Delta x_{o,j}}\right)^T \left(\frac{\Delta x_i}{\Delta x_{o,j}}\right)$, whose largest eigenvalue $\sigma$
is connected to
$\lambda = \frac{1}{T} \ln \sqrt{\sigma}$ .                                                                          (1)
Here $\Delta x_{0,i}$ and $\Delta x_i$ are the initial and final distance in the $i^{\text{th}}$-direction between neighboring
trajectories. This algorithm requires dense regularly-spaced orthogonal grids of trajectories. For



the SPLASH dataset, we manually chose quadruplets of 4 neighboring trajectories that form a
near-rectangle, define the local orthogonal coorditate system most strongly aligned with the axes
of the near-rectangle, and then estimate FTLEs using eq. (1) for the center of mass of each
quadruplet (Fig. 2 shows the quadruplets and their centers of mass locations).
A modification for unstructured meshes was described in Lekien and Ross (2010), where FTLE
for each trajectory is estimated using its $N$ closest neighbors as
$\lambda = \frac{1}{T} \ln \tilde{\sigma}$ ,                                    (2)
where $\tilde{\sigma}$ is the largest positive singular value of a matrix
$\tilde{M} = DX_f \, (DX_0)^T \, (DX_0(DX_0)^T)^{-1}$
which minimizes $\left\| DX_f - M \, DX_0 \right\|$.
Here $DX_0 = \begin{pmatrix} x_1^0 - x_i^0 & ... & x_N^0 - x_i^0 \\ y_1^0 - y_i^0 & ... & y_N^0 - y_i^0 \\ z_1^0 - z_i^0 & ... & z_N^0 - z_i^0 \end{pmatrix}$ and $DX_f = \begin{pmatrix} x_1^f - x_i^f & ... & x_N^f - x_i^f \\ y_1^f - y_i^f & ... & y_N^f - y_i^f \\ z_1^f - z_i^f & ... & z_N^f - z_i^f \end{pmatrix}$ are matrices of the
initial and final displacements between the trajectory and its N neighbors. Because the largest
singular value of $\tilde{M}$ is equal to the square root of the largest eigenvalue of $\tilde{G} = \tilde{M}^T \tilde{M}$, eq. (2) is
the unstructured-mesh counterpart of eq. (1). We use the Delaunay triangulation partition to
define closest neighbors for each drifter (Fig. 2 shows the Delaunay partition for the SPLASH
dataset). The FTLE is then estimated using eq. (2) at each drifter's initial position using its
Delaunay closest neighbors. When used together, a combination of these two methods – the
regular and the unstructured mesh methods – allows estimating FTLEs both at the locations of
each drifter and between neighboring quadruplets.



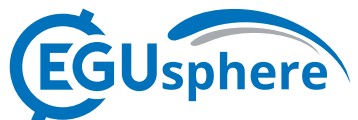


**b) Trajectory path length**

Trajectory path length $L = \int ds = \int_{t_0}^{t_0+T} |\vec{u}(x(t), t)| dt$, where $ds$ is the incremental length of
the infinitesimal trajectory segments. For drifter data, summation can be used instead of
integration. $L$ has been proposed by Rypina et al. (2011) as one of the "Trajectory Complexity
measures" and by Mancho et al. (2013) as the "Lagrangian Descriptor" for identifying LCSs.
Curves of near-constant $L$ values with a large $\nabla L$ in the perpendicular direction to the curve are
indicative of the stable manifolds of hyperbolic trajectories (because trajectories on the manifold
approach the hyperbolic trajectory and trajectories slightly off the manifold are repelled from it).
This method is less mathematically rigorous than FTLEs and is frame-dependence, but it is
commonly used due to its simplicity.

**c) Trajectory correlation dimension**

Trajectory correlation dimension ($C$) is a measure of space occupied by a trajectory. In 2D, it
varies from 0 for a point, to 1 for a curve, to 2 for a trajectory that densely fills an area. $C$ can be
estimated using a box counting algorithm, where the entire trajectory data set is first mapped
onto a unit square, and the unit square is then repeatedly split into $2^{-2m}$, $m = 0, 1, ..., M$ adjacent
square boxes with side length $s = 2^{-m}$ (we use $M = 12$ in this paper). A distribution function is
then computed for each trajectory as $F_i(s) = \frac{1}{N_i^2} \Sigma \left( N_i^j(s) \right)^2$ where $N_i$ is the total number of
points in the $i^{th}$ trajectory and $N_i^j$ is the number of points in the $i^{th}$ trajectory that fall inside the $j^{th}$
box for a given $s$. The trajectory correlation dimension $C_i$ for the $i^{th}$ trajectory can then be
estimated as the slope of $F_i(s)$ vs. $s$ in log-log coordinates. Just like trajectory path length, $C$ is





another measure of "Trajectory Complexity" and has been proposed by Rypina et al. (2011) as a
means for LCS identification. Similar to $L$, level curves of near-constant $C$ with a large $\nabla C$ in the
perpendicular direction to the curve are indicative of the stable manifolds of hyperbolic
trajectories. $C$ is a more sensitive measure of "Trajectory Complexity" than $L$ but is more
computationally expensive. Just like $L$, $C$ is also frame dependent.

**d) Trajectory encounter number and trajectory encounter volume**
Trajectory encounter volume $V_{en}$ for a particular trajectory is a volume of fluid that gets in
contact with a particular water parcel over a time interval $T$ (Rypina and Pratt, 2017; Rypina et
al., 2018). This is a frame-independent quantity. It quantifies the mixing potential of a flow and
is related to the eddy or turbulent flow diffusivity $\kappa$ (Rypina et al., 2018). The larger $V_{en}$, the
more opportunities exist for a parcel to exchange properties with surrounding fluid. Smallest $V_{en}$
occur in isolated secluded regions of the flow such as eddy cores, and largest $V_{en}$ occur in
hyperbolic regions and along the stable manifolds of hyperbolic trajectories. Thus, $V_{en}$ can be
used to characterize both elliptic and hyperbolic LCSs.
For data sets containing a finite number of particle trajectories, encounter volume for a particular
trajectory can be approximated by assigning small volumes $\delta V_j$ to all trajectories and summing
over those trajectories that come close to the particular trajectory: $V_{en} \approx \sum \delta V_j$. For regular grids
$\delta V_j = \delta V = \ const$ and $V_{en} = \delta V N_{en}$ where $N_{en}$ is the encounter number – the number of
trajectories that come close (i.e., within a small radius $R$) to the particular trajectory. In our
calculations, we use $R = 1$ km and $\delta V \approx 1$ km$^2$, which is the square of the mean distance
between the drifters' release locations.

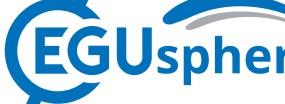

**e) Dilation**
Dilation is the velocity divergence averaged along a particle's trajectory,
$D = \frac{1}{T} \int_{t_0}^{t_0+T} div\big(u(x(t), t)\big)$. This frame-independent quantity was proposed by Huntley et al.
(2015) as a method for identifying clusters of material at the ocean surface. Trajectories with the
largest positive/negative $D$ experience the strongest divergence/convergence and thus
repel/accumulate buoyant floating surface tracers (including drifters). For drifter data,
summation can be used instead of integration, and the Linear Least Squares method of Molinari
and Kirwan (1975) can be used to estimate $div(u)$ at each point along each trajectory.

**f) Lagrangian-Averaged Vorticity Deviation (LAVD)**
$LAVD$ is the vorticity deviation with respect to the domain-averaged instantaneous vorticity,
averaged along a particle trajectory, $LAVD = \frac{1}{T} \int_{t_0}^{t_0+T} |\omega(x(x_0, t) - \overline{\omega(t)}| \, dt$. It was introduced
by Haller et al. (2016) as a frame-independent metric for identifying rotationally coherent
Lagrangian eddies, which correspond to a region contained within the outermost convex level
surface of $LAVD$ surrounding a maximum. For drifter data, we again use summation instead of
integration and estimate vorticity using a Linear Least Squares method.

**g) Spectral Clustering**
The last method for identifying the LCSs that we will be testing using drifter data is the
optimized-parameter Spectral Clustering described in Filippi et al., 2021a;b (see also Shi and
Malik (2000), Hadjighasem et al. (2016) and references therein). This method aims at
identifying, within a given dataset of trajectories, clusters of trajectories that are most similar to





each other and, at the same time, most dissimilar from trajectories in other clusters. The method
starts with the construction of a matrix of weights
$w_{ij} = \begin{cases} \dfrac{1}{r_{ij}} \\ w_{diag} \end{cases}$ , where $r_{ij}$ is the time-average distance between $i^{th}$ and $j^{th}$ trajectory, and $w_{diag}$ is
a large constant offset value (we use $w_{diag} = max(w_{ij}) \times 10^7$). Based on this matrix, the
method used ideas from machine learning theory, specifically, N-cut matrix partitioning and K-
means clustering algorithms, to identify the spectral clusters with the largest/smallest degree of
intra-/inter-cluster similarity. Importantly, the optimized-parameter version of the Spectral
Clustering method (Filippi et al., 2021a;b) that we are using automatically detects both the
optimal number of clusters and the clusters sizes (based on the normalized eigengap between the
eigenvalues of the generalized normalized Laplacian, as described in Filippi et al. (2021a)).
Being based on the distances between trajectories, spectral clustering is frame-independent.

**h) Linear Least Squares (LLS) method for estimating drifter-based divergence and**
**vorticity**
In order to estimate divergence and vorticity from drifters, we follow the approach of Rypina et
al. (2021), where we first compute horizontal velocities from drifter positions using a centered
finite-difference scheme and then apply the linear least squares (LLS) method of Molinari and
Kirwan (1975) to estimate horizontal velocity gradients. LLS method is based on the Taylor
expansion of velocity, $U = DA$, where $U = [u_1, \dots u_N]$ is a (known) vector containing the
$u-$velocity at a given time $t$ for each of the $N$ drifters, $D = \begin{pmatrix} 1 & x_1 - \bar{x} & y_1 - \bar{y} \\ \vdots & \vdots & \vdots \\ 1 & x_N - \bar{x} & y_N - \bar{y} \end{pmatrix}$ is a known
distance matrix containing instantaneous distances from each drifter to the center of mass of the



drifter distribution at time $t$, and $A = \left[\bar{u}, \frac{\partial u}{\partial x}, \frac{\partial u}{\partial y}\right]$ is the vector containing the unknown velocity
derivatives at time $t$ that can be estimated using the Moore-Penrose pseudo-inverse as $A =$
$(D^T D)^{-1} D^T U$ (and similarly for the v-component). The accuracy of the LLS method increases
with increasing number of drifters and decreasing aspect ratio of the drifter configuration, i.e.,
LLS works best for tight equidistant polygons with many drifters. Following Essink et al. (2021),
we will refer to the LLS estimates as trustworthy (and mark them by colored circles) if there are
$\geq 6$ drifters within a $3 - \mathrm{km}$ radius, the center of mass of the drifter distribution is located within
the polygon, and the polygon aspect ratio is $\leq 6$. If the aspect ratio condition is not satisfied (but
the number of drifters, the distance, and the center of mass conditions are), we still compute LLS
estimates but we refer to them as less trustworthy (and mark them by colored diamonds). In all
other cases, we do not produce estimates of divergence and vorticity. Note that methods other
than LLC can also be used to compute divergence and vorticity: divergence can be estimated as a
rate of change of the area spanned by the drifter polygon, and both divergence and vorticity can
be estimated using Green's theorem as, respectively, the circulation around and total flux
through the drifter polygon. Rypina et al. (2021) compared all three techniques in detail using
both real and simulated drifters deployed at similar inter-drifter distances as the SPLASH drifters
and observed good correspondence between all three techniques for clusters of 6 drifters as long
as the drifters stayed within a few km of each other and the aspect ratio was reasonably small
($\leq 5$). Because the LLC method was yielding slightly less noisy results for marginal inter-drifter
distances and aspect ratios, here we show results for the LLS rather than other techniques.

**3. Results**



We start by qualitatively separating the motion of drifters into three stages. For about a day after
deployment, all drifters started moving together in an anticyclonic fashion to the north and then
northeast towards the coast (Fig. 1) – this is what we will refer to as the initial stage of motion.
Upon approaching the shelf, the drifters halted their on-shore motion and split into two groups, a
smaller northern group that headed northward along the coast and a larger southern group that
moved southward. This splitting behavior was reminiscent of a hyperbolic motion in the vicinity
of a hyperbolic trajectory, with a stable manifold emanating from a hyperbolic trajectory in the
off-shore direction, and two unstable manifolds northward and southward from it in the along
shore direction. As a result, a long and narrow filament roughly aligned with the coast is quickly
formed just after 1 day. This filament contains about one third of all the drifters, with the rest of
the drifters forming a less elongated and more compact blob just south-southwest of the filament.
Some clustering of temporarily occurs at about 1 day near the southeastern corner of the drifter
configuration but goes away later. The slow-down of the on-shore movement, the splitting into
the north-south groups, and the formation of the elongated along-shore filament constitute the
second stage of motion, which lasted from about 0.9 to about 1.25 days after the deployment.
Finally, during the third stage of motion, the drifters started moving off-shore to the southwest.
As they progress further from the coast, trajectories started exhibiting more looping and the
drifters dispersed further apart from one another, although they still remained in an elongated
filament configuration (not anymore aligned with the coast) all the way until day 5, which is the
end time of this dataset.
Having split the drifter movement into 3 stages, we next apply our Lagrangian methods to
trajectory segments from $t_{start} = 0$ days until $t_{end} = 0.5, 1,$ and 3 days, respectively (top,
middle, and bottom row of panels in Figs. 3-9). The resulting fields highlight the dominant LCSs

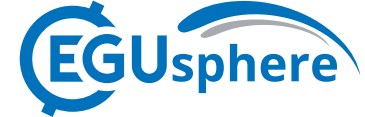

that existed at the time of the drifter deployment (i.e., at $t_{start} = 0$) and that governed the
movement of drifters during the subsequent 0.5, 1, and 3 days, respectively. (A movie of the
FTLEs at $t_{start} = 0$ computed with progressively increasing $t_{end}$ is included in the
Supplementary Material.)
*FTLEs (Fig. 3):* During the initial stage of motion ($t_{start} = 0$ days and $t_{end} = 0.5$ days),
FTLEs were generally smaller for drifters released in the north and south, and larger in between,
especially in the west. As evident from the upper middle panel of Fig. 4 (FTLEs mapped to the
current position of drifters on day 0.5), this behavior was dominated by the slightly larger
separations (empty spaces) between drifters in the middle of the distribution and slightly tighter
drifter distributions on the east and west sides. Later on, this small inhomogeneity in the early-
time drifter separations became less important, and was overrun by other features. Specifically,
during the intermediate stage ($t_{start} = 0$ days and $t_{end} = 1$ day), the largest FTLEs were
observed along the northwestern edge of the release domain, containing drifters that split north-
south upon approaching the coast and formed an elongated along-shelf filament. FTLEs were
negative for drifters released near the middle of the northeastern edge of the release domain,
which converged into a tight cluster in the southeastern corner of the drifter distribution at 1
day. This feature was transient and disappeared as the drifters moved offshore. The rest of the
release domain has small positive FTLE values; these were the drifters which did not experience
strong along-shore alignment and formed a more compact group in the northern part of the
drifter distribution at 1 day. Finally, during the third stage of motion ($t_{start} = 0$ days and
$t_{end} = 3$ days), as the drifters moved offshore and re-shaped into a northwest-southeast
configuration, the only distinguishing feature of the FTLE field was the blue cluster near the
central part of the release domain. This cluster contained trajectories that remained closer to each



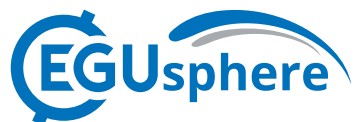

other than other trajectories. When mapped to the current positions of the drifters at *3* days, these
smallest blue FTLEs corresponded to a group of drifters in the western part of the distribution,
i.e., a cluster of blue dots in the lower middle and right panels of Fig. 4. (Note that the northwest-
southeast configuration at 3 days was mostly formed from the drifters located in the southern
part of the distribution at 1 day, and so is different from the along-shelf "tail".)
Overall, at all times the FTLE fields showed pronounced variations in values across the drifter
distribution. Largest FTLEs indicated regions of strong drifter separation that, during the
intermediate stage of motion, were reminiscent of stable manifolds of hyperbolic trajectories.
Smallest FTLEs highlighted groups of drifters that stayed closer together compared to their
neighbors. Transient negative FTLE regions were also present and highlighted groups of drifters
temporarily converging into tight clusters (before spreading apart again later on). The FTLEs
varied significantly with the increasing duration of trajectories, i.e., increasing $t_{end}$, suggesting
that different flow features governed the movement of drifters during different stages of motion.
The calculation of FTLEs was straightforward and computationally inexpensive, and by
combining the structured and unstructured grid methods, we were able to obtain FTLE values at
both drifter release positions and in between them, providing twice higher resolution compared
to other methods.

*L (Fig. 4):* Trajectory path length *L* showed a monotonic increase in values with increasing
latitude across the release domain at all times, with the largest/smallest values in the
northwest/southeast. This large-scale gradient in *L* was dominated by the faster anticyclonic
motion of the northwestern drifters at early times. This was reminiscent of a solid body rotation,
where the northwestern drifters that were located further from the center of rotation than their



southeastern neighbors moved at a faster speed and thus covered a longer path length over a
given time interval. (This effect could presumably be removed by recalculation of $L$ in an
appropriate rotating frame of reference, an operation that would not change the values of the
FTLEs. Thus it is perhaps not surprising that the distributions of the two metrics differ in
significant ways.) All other characteristic features, such as the splitting of trajectories into the
northern and southern group at about 1 day, the formation of an elongated along-shelf filament,
the transient convergence region, and the reshaping of the drifter configuration as it progressed
further offshore had only minor effects on the resulting path length fields. We also looked at the
gradient of the path length field as a means of identifying LCSs, but the gradient was noisy and
did not yield any distinguished maxima. Thus, despite being easy to compute and straightforward
to interpret, the path length $L$ was only marginally useful in identifying the dominant LCSs.

*CD (Fig. 5):* Results for the trajectory correlation dimension $CD$ were generally similar to those
for the trajectory path length, in that $CD$ was also dominated by the across domain gradient from
northwest to southeast, and the distribution of $CD$ did not change dramatically in time. Being a
more sensitive measure of the trajectory complexity than $L$, $CD$ exhibited a slightly stronger
variability across the domain. This sensitivity of $CD$ came at a cost of being more
computationally expensive than $L$, and, in the end, was still not enough to identify the LCSs
responsible for neither the formation of the elongated filament at 1 day, nor the transient
convergence zones just after 1 day, nor the suppressed separation between trajectories coming
from the central part of the domain at 3 days. Overall, $CD$ was no more useful than $L$ in
identifying the LCS, and, like $L$, had the same frame dependence issues.



$V_{en}$ ($Fig. 6$): The encounter volume $V_{en}$ showed a lot of variability across the domain and was
able to successfully highlight several different flow features governing the movement of drifters
at different stages of motion. During the initial stage, $V_{en}$ had largest values in the southern part
of the release domain. From the top middle panel (the map of $V_{en}$ at the current position of the
drifters) we observed that these enhanced values were caused by the tighter clustering of drifters
(so that they were able to meet more neighbors). During the intermediate stage of motion, the
distribution of $V_{en}$ changed, and the largest values migrated to the northeastern edge of the
release domain. This was associated with the transient convergence zone (that we also observed
in the FTLE fields); trajectories released in that area converged into a tight cluster located at the
southeastern corner of the drifter distribution at $1$ day ($2^{nd}$ row, middle panel). The elongated
along-shore filament seen at $1$ day contained smallest $V_{en}$ values since trajectories in the
filament did not encounter many neighbors. Trajectories that headed north after approaching the
coast at $1$ day never caught up with the rest of the distribution, always staying behind, i.e., to the
north from the rest of the drifters. Thus, these drifters experienced the least amount of encounters
and, during the third stage of motion, had the smallest $V_{en}$ values. Apart from this low-
encounter-number group, there were no other pronounced features in the $V_{en}$ field during the
third stage of motion.
It is interesting to compare and contrast $V_{en}$ with FTLEs, which became sort of a benchmark for
the LCS detection problems, being frame independent, commonly used, and easy to compute.
There are significant differences between the distributions of the two metrics, reflecting
differences in what the two are actually measuring. While both FTLEs and $V_{en}$ are sensitive to
flow convergence/divergence, trajectory clustering, and hyperbolic behavior, one of the key
differences between them is that $V_{en}$ is a time-integrated measure that depends on the behavior of





trajectories over the entire time interval between the initial and final times, whereas FTLEs only
depend on the initial and final positions of drifters (i.e., FTLEs do not care how trajectories got
to their final positions, whereas $V_{en}$ does). For example, even though trajectories comprising the
low-FTLE blue cluster in the western part of the distribution at 3 days have come close together
at that time, over a time frame of 3 days they experienced no more trajectory encounters than
many other trajectories outside of that blue FTLE cluster (and thus were not standing out in the
$V_{en}$ field).
Overall, the encounter volume $V_{en}$ proved to be an interesting frame-independent diagnostic that
was sensitive to both enhanced clustering, hyperbolic behavior, and flow convergence, and was
complementary to FTLEs.

*D (Fig. 7):* The challenge with computing dilation $D$ (as well as $LAVD$) for real drifters is the
inability to reliably estimate divergence (vorticity) for isolated drifters and drifters forming
strongly elongated polygons. This was not a problem for SPLASH drifters during the early stage
of motion but became an issue as the drifters started to spread apart and formed elongated
filaments. During the initial stage of motion (top row), the most pronounced feature of the $D$
field was the negative cluster in the southern corner of the release domain, which contained
drifters that converged more than their neighbors. A similar feature has been identified by $V_{en}$ as
the high-encounter-volume region. The rest of the domain had near-zero dilation. During the
second stage of motion (middle row), the negative dilation in the south diminished, and another
convergent negative-$D$ region appeared along the northeastern edge and eastern corner of the
release domain. This is reminiscent of the negative-FTLE / high-$V_{en}$ region in the middle rows of
Figs. 3 and 6. Trajectories released there converged into the southeastern corner of the drifter



distribution at *1* day. Around this time, an increasing number of trajectories started having
unreliable divergence values; for example, divergence and thus dilation, could not anymore be
reliably computed for the northern group of trajectories, which became too few and too sparse.
During the third stage of motion, this problem became even more important and by day *3*, the
dilation field was undefined for about half of the trajectories. The resulting *D* field was noisy and
did not exhibit any pronounced features.
Overall, dilation *D* was useful in highlighting the convergence zones during the first two stages
of motion, but numerical difficulties associated with reliably estimating divergence for sparse
datasets and elongated drifter configurations made it challenging to compute *D* over long time
intervals from real drifters.

*LAVD (Fig. 8):* During the first stage of motion, the strongest feature in the *LAVD* map was the
red large-*LAVD* region near the southern corner of the release domain. This area coincided
roughly with the negative-*D* and large-$V_{en}$ in Figs. 6-7. During the second stage of motion, this
feature diminished in intensity and a second high-*LAVD* region appeared near the eastern corner
of the domain. Again, a similar region has been highlighted by low FTLEs, high $V_{en}$, and
negative *D*, although *LAVD* emphasized the eastern corner rather than the entire northeastern
edge of the release domain. Trajectories starting there converged into a tight cluster near the
southeastern corner of the drifter distribution at *1* day.  It is interesting that *LAVD* identified
similar regions as FTLEs, $V_{en}$, and *D*, despite the fact that clustering behavior and flow
convergence do not necessarily need to be associated with increased vorticity deviation. In our
case, clustering and convergence did coincide with increased vorticity deviation, suggesting that
perhaps a small-scale eddy or recirculation that was affecting this particular cluster of drifters



might have been responsible for all of these effects. (Note that interpreting the vorticity deviation
as vorticity is only possible when the domain-averaged background vorticity, $\bar{\omega}$ , is small, which
was not always the case for the SPLASH drifters.) Finally, during the third stage of motion
(bottom row), the map of $LAVD$ became gappy (because, similar to the challenges with dilation,
here we could not reliably estimate $LAVD$ for about half of the drifters) and showed no
distinguished regions. However, when mapped to the current position of the drifters (lower
middle panel), the cluster in the middle of the drifter distribution showed larger $LAVD$ values
than clusters to the northwest and southeast (but since trajectories forming the middle cluster
came from different parts of the release domain, this feature did not stand out in the left panel).
Overall, during the first two stages of motion, $LAVD$ highlighted two regions with enhanced
$LAVD$ values. While large $LAVD$ does not generally indicate convergence, in our case both
regions were strongly convergent. At later times, vorticity estimation became less reliable, and it
became harder to distinguish coherent features in the sparse and noisy map of $LAVD$. Note that
our high-$LAVD$ regions differed from the classical examples of coherent rotational eddies. Our
regions were not circular, did not have a single maximum, and were too noisy to identify the
outermost convex contour level (which marks the outer edge of the coherent rotational eddies in
the standard application of the $LAVD$ technique). Thus we cannot call these high-$LAVD$ features
Lagrangian coherent eddies.

*Spectral Clustering (Fig 9):* At early times, the number of coherent clusters identified by the SC
algorithm was quite large (12), although some clusters only contained a few drifters. (Recall that
the optimized-parameter SC is able to autonomously identify the optimal number and optimal
size of the clusters, without input from the user). Among the detected clusters, the yellow cluster



located in the south-southwest of the release domain is perhaps the most noteworthy because it
resembled the low-FTLE / large-$V_{en}$/ negative-$D$/ large-$LAVD$ region that contained trajectories
that stayed close together during the initial stage of motion. As the drifters entered the second
stage of motion, the number of identified coherent clusters decreased to 6. Most of the release
domain was split between two large clusters – the cyan cluster in the north-northeast containing
drifters attracted by the convergence region (i.e., drifters that converged/came close to the
southeastern corner of the drifter distribution at 1 day), and the green cluster in the south of the
release domain containing drifters that did not feel the pull of that convergence zone. The
remainder of the domain, i.e., the northwestern edge of the domain that mostly contained the
trajectories forming an elongated along-shore filament, was split into 4 more clusters. Finally, at
the third stage of motion, the drifters were split into 8 clusters, and the grouping was most
straightforward to interpret by looking at the lower middle panel. All trajectories in the western
cluster were blue (these trajectories came from the central and southern portion of the domain in
the bottom left panel), with the yellow cluster to the southeast of it (these trajectories came from
around the periphery of the blue cluster in the bottom left panel), and with the orange group
further to the southeast of the yellow cluster (most orange trajectories originate from the
northeastern edge of the release domain in the bottom left panel). The remaining 5 clusters only
contained 1 or 2 trajectories.
Overall, the spectral clustering algorithm seems to have identified physically-meaningful and
intuitively-clear coherent clusters; the movement was similar for drifters within each cluster and
dissimilar between the clusters. There were also good correspondences between the spectral
clusters and coherent featured highlighted by other methods.





## 4. Summary and Discussion

SPLASH drifter experiment provided the long-awaited opportunity to test the performance of different dynamical systems techniques with real, rather than simulated, ocean drifters. Although many other drifter data sets are available for various regions of the World Ocean, drifters are typically released by a handful here and there, and the resulting data is typically inadequate for mapping out the LCS. For example, the NOAA's Global Drifter Program data set contains several thousands of near-surface drifter trajectories released between 1971 and today, but the density of the drifter distribution at any given time is only about 1 per 5-by-5 deg box, which is too sparse to identify even mesoscale LCSs.

Three qualitatively-different stages of motion were evident in the SPLASH drifter data. During the first stage, all drifters moved anticyclonically toward the coast. During the second stage, the drifters halted their on-shore motion, split north-south, and formed an elongated along-shelf filament. During the third stage, the drifters moved off-shore, rearranging themselves into a northwest-southeast configuration. As the character of drifter movement changed with time, the maps of the Lagrangian metrics and the resulting LCSs that they highlighted changed as well. In order to capture this time-dependence, we have applied the Lagrangian metrics to segments of trajectories from fixed $t_{start} = 0$ days to variable $t_{end} = 0.5, 1$, and $3$ days, respectively. When the Lagrangian metrics were mapped back to the initial positions of drifters at $t_{start}$, the resulting maps highlighted the dominant LCS which existed at the time of the deployment within the deployment domain, and which govern the subsequent motion of drifters over the corresponding time interval. The fact that the results for any particular measure differed between the three time intervals is consistent with submesoscale dynamics, where fronts, small eddies, and filaments form, evolve, and disappear on time scales of days or less.





The Lagrangian techniques we have examined include FTLEs, trajectory path length, trajectory
correlation dimension, trajectory encounter number, dilation, $LAVD$, and optimized-parameter
spectral clustering. This list was motivated by Hadjighasem et all. (2017) and is by no means
exhaustive, but it includes a variety of commonly-used methods that are based on different
properties of trajectories, make use of the different definitions of coherence, and thus aim to
identify different types of LCSs. Interestingly, despite the differences in their underlying
principles and methodologies, many of these methods identified similar features within the
SPLASH drifter data set.
Among the most prominent features that were highlighted by multiple methods were: 1) the
region near the northwestern edge of the release domain (red FTLEs, blue $V_{en}$, a group of
spectral clusters other than dark blue), which contained trajectories that split north-south upon
approaching the shelf and formed an elongated along-shelf filament at about 1 day; 2) the very
strong but transient convergence region located near the northeastern edge of the release domain
(negative FTLEs, large $V_{en}$, strongly negative dilation, cyan spectral cluster), which contained
trajectories that converged into a tight cluster at about 1 day; and 3) the region in the
central/southern part of the release domain (small FTLEs, blue spectral cluster), which contained
trajectories that remained close to each other starting from 2.5 days and onward.
Although all of the identified structures were noisier and more complex than the classical elliptic
and hyperbolic LCSs in textbook examples, some of the features bore resemblance to their
classical counterparts. For example, the north-south splitting of trajectories starting within the
red FTLE region near the northwestern edge of the domain was qualitatively similar to the
behavior of trajectories near a hyperbolic region, where particles approach the hyperbolic
trajectory along a stable manifold and then split and move away from the hyperbolic trajectory



along the two unstable directions. The detected large-FTLE region near the northwestern edge of
the release domain might thus possibly indicate the presence of a stable manifold in this region.
From the standpoint of numerical efficiency, FTLEs and $L$ were the least computationally
expensive, whereas $CD$, $V_{en}$, and Spectral Clustering were the most computationally expensive.
However, with only 135 trajectories, the differences in the amount of time required to apply each
technique were not critical. More importantly, FTLEs had the advantage of providing values at
the positions of each drifter as well as between the neighboring drifters, effectively yielding
output fields with twice the resolution of the other methods. FTLEs were also less affected by the
gaps in GPS transmissions along trajectories, because the estimation of FTLEs at a particular
time only required knowing the initial and the current positions of the drifters, rather than
requiring the information about the entire trajectory up to that time, as in the case of all other
methods - path length, correlation dimension, encounter number, dilation, $LAVD$, and spectral
clusters.
It is interesting to note that the two frame-dependent methods – $L$ and $CD$, which were
dominated by the large-scale gradient across the entire release domain and did not highlight any
submesoscale features – were the least useful in identifying LCSs.
Massive drifter releases such as SPLASH are extremely useful for improving our understanding
of the transport and exchange processes at submesoscale. Specifically, data from the SPLASH
and other similar experiments have been used for estimating diffusivity and studying particle
spreading regimes at submesoscale (Poje et al., 2014; Beron-Vera and LaCasce, 2016). We have
shown that a simultaneous release of about 100 drifters provides a glimpse of the dominant
Lagrangian Coherent Structures that govern the transport of water and the movement of drifters.



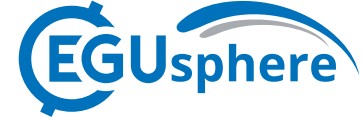

SPLASH experiment was not specifically focused on identifying LCSs, so the drifter release
locations and timings were not optimized for capturing the underlying LCSs. Our analysis
suggested that, luckily, a stable manifold of a hyperbolic trajectory was likely present in the
northwestern edge of the domain spanned by the drifters at the time of their release and persisted
for at least the first 1-1.5 days of the experiment. As explained above, this feature manifested
itself as a high-FTLE region and was characterized by the north-south splitting of trajectories
around day 1. However, no clear elliptic LCSs (i.e., coherent eddy cores) were identified by any
of the methods, even though an anticyclone was likely present in the region (based on the
numerical model simulations and the clockwise movement of drifters during the first day after
release). It is difficult to say whether this anticyclone did not possess a Lagrangian core, or the
core was located outside of the drifter release domain, or whether the core was not properly
resolved by the SPLASH drifters. In the future, it would be interesting to repeat the experiment
with the drifter deployment site and the release pattern optimized for capturing specific LCSs
whose presence could have been predicted based on a model or satellite data.
The very rapid nature of evolution at submesoscales may cause an evenly spaced array of drifters
to rapidly collect into filaments, making it difficult to continue to accurately compute certain
Lagrangian measures.  In SPLASH experiment, for example, the nearly-rectangular deployment
mesh of drifters (which took quite a bit of effort to achieve) eroded into an elongated filament
over a time scale of about a day. Note, however, that it is precisely this rapid filamentation
process and the rapid deformation of the initial mesh that gives rise to the strong, pronounced,
and detectable LCSs.
Finally, in order to investigate the robustness and reliability of the real-drifter-derived LCS, we
have simulated the SPLASH drifter dataset in a model and then compared the resulting



SPLASH-like drifter-based LCS to those computed using dense regular orthogonal grids of
trajectories (we refer to the latter as true model fields). We used the operational data-assimilative
Navy Coastal Ocean Model (NCOM) forecasting model for this purposes
(https://data.gulfresearchinitiative.org/data/R4.x265.245:0002). Comparison between SPLASH-
like and true model fields showed good agreement, the Lagrangian metrics for the real and
simulated SPLASH drifter were of the same magnitude, and the overall geometries and types of
the coherent features were similar in the model and in the real ocean, suggesting that SPLASH-
based fields were robust and reliable. Detailed comparisons can be found in Supplementary
Material.
**Acknowledgements:** Many thanks to Gregg Jacobs of NRL Stennis and William Nichols at
GRIIDC for sharing the output of the NCOM model. We are grateful to Margaux Filippi and
Alireza Hadjigjasem for their help with the spectral clustering code. IR and LP would like to
acknowledge support from the ONR CALYPSO grant #N000141812417. TG is thankful to ONR
for supporting his NAVY Master's program fellowship at WHOI. TO was supported by ONR
CALYPSO grant #N000141812138.
**Data Availability Statement**
Data from the Submesoscale Processes and Lagrangian Analysis on the Shelf (SPLASH) surface
drifters used in this paper is available from:
https://data.gulfresearchinitiative.org/data/R4.x265.000:0074.
The NCOM model output fields used in this paper are available from:
https://data.gulfresearchinitiative.org/data/R4.x265.245:0002.



**Author Contribution Statement:** IR led the overall effort and primarily wrote the manuscript,
TG performed estimation of most Lagrangian metrics, LP and TO contributed to the
interpretation of the results and editing of the manuscript.
**Competing interests Statement:** no competing interests














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

**Figure 1. Trajectories of the SPLASH drifters, with their positions at the release time, 1 day, and 3 days shown by black, blue,**

**and green dots, respectively. The inset shows the geographical location of the experiment cite, with black box indicating the**

**domain shown in the main panel.**










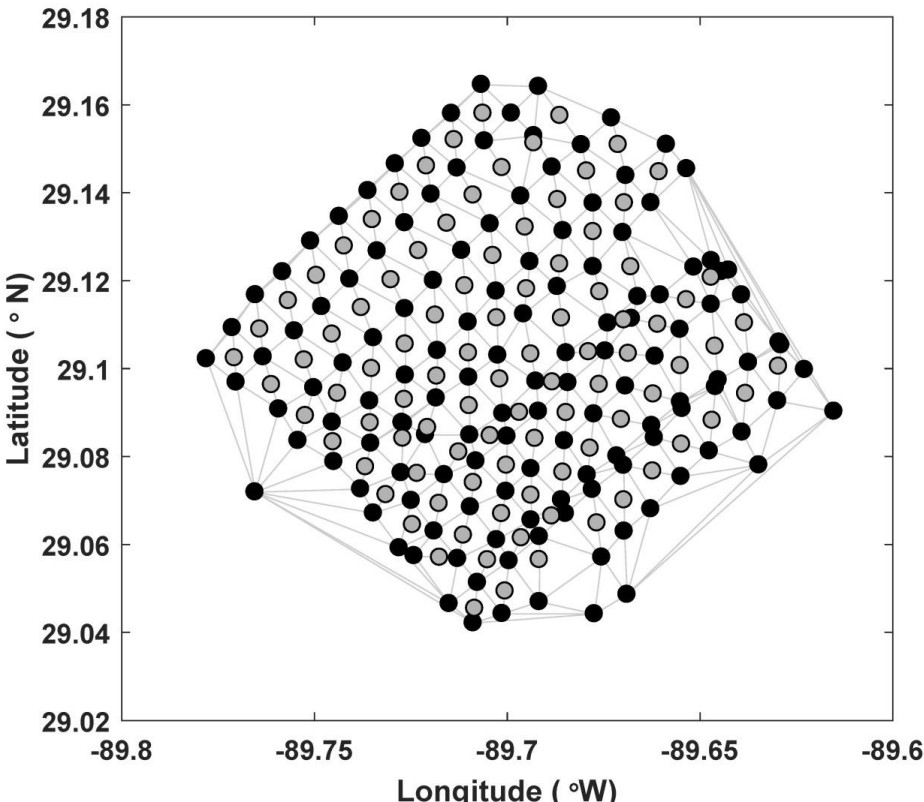


**Figure 2. Release locations of the SPLASH drifters (black dots) with the Delaunay delineation (grey lines) used to define the closest neighbors for estimating FTLEs at the drifter positions using the unstructured grid method. Grey circles show locations between the drifters, at which FTLEs were estimated via the structured grid method (using a quadruplet of black drifters around each grey dot).**













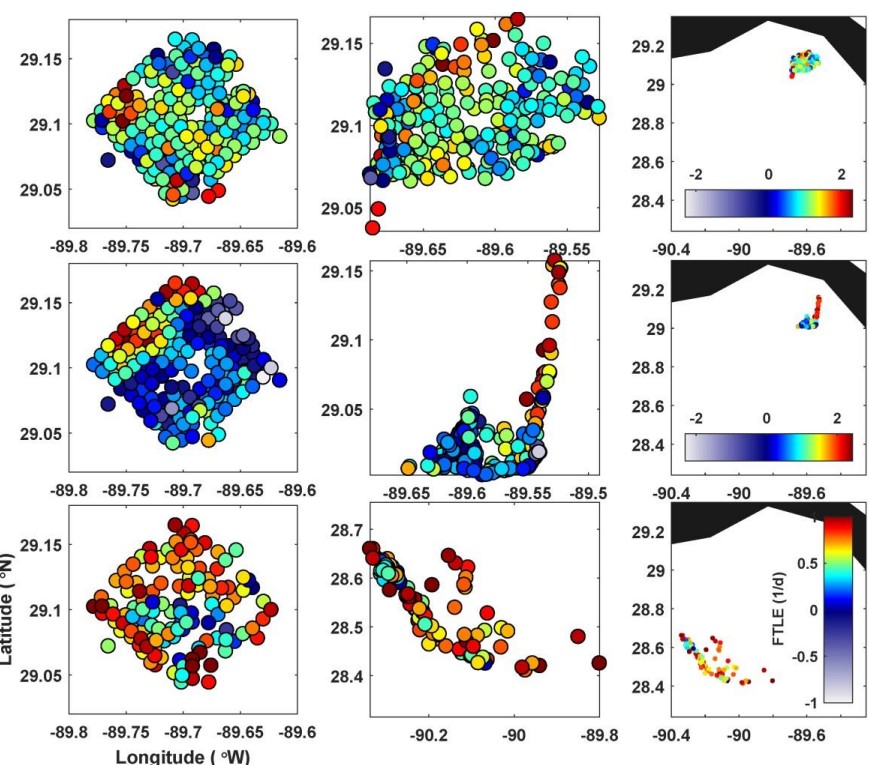


**Figure 3. Real-drifter-based FTLEs at (top) 0.5, (middle) 1, and (bottom) 2 days, mapped to the initial (left) and current**
**(middle and right) positions of the drifters.**



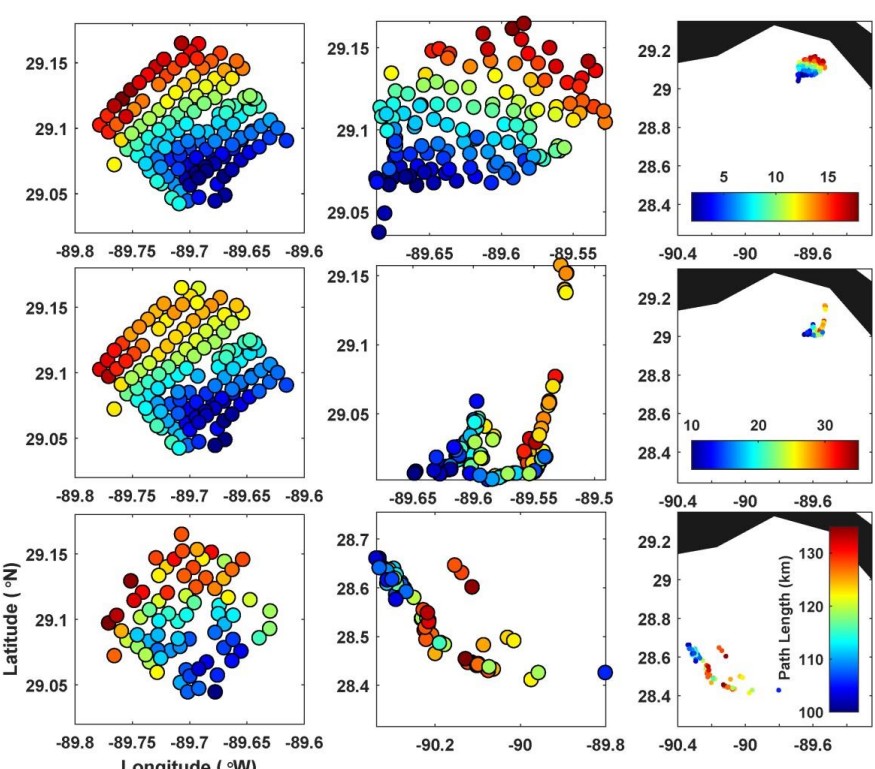


**Figure 4. Real-drifter-based path length at (top) 0.5, (middle) 1, and (bottom) 3 days, mapped to the initial (left) and current**

**(middle and right) positions of drifters.**






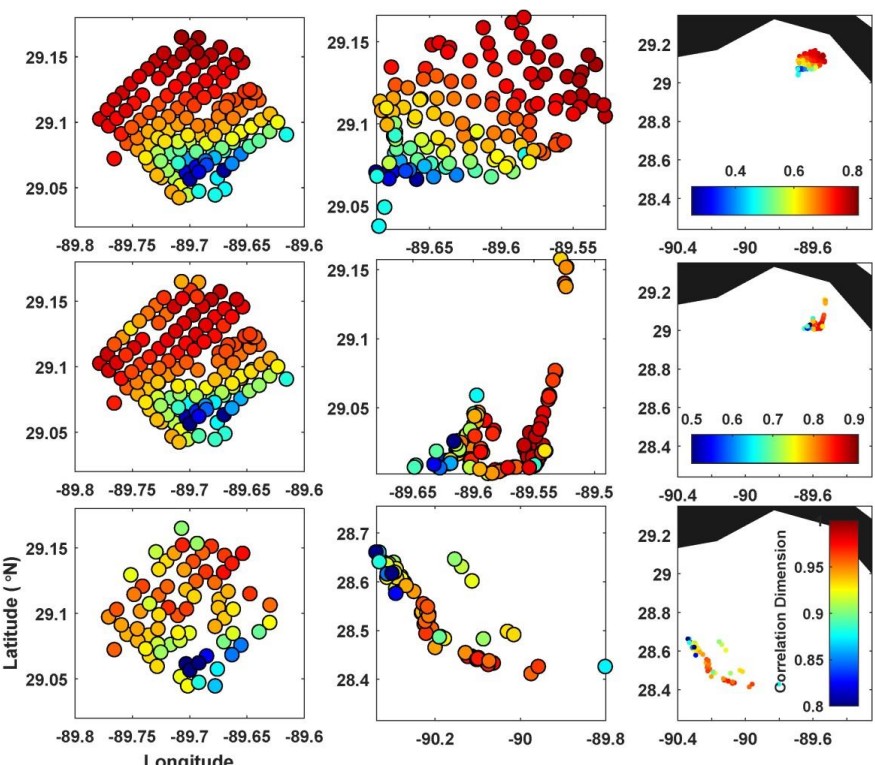

**Figure 5. Real-drifter-based correlation dimension at (top) 0.5, (middle) 1, and (bottom) 3 days, mapped to the initial (left) and current (middle and right) positions of drifters.**





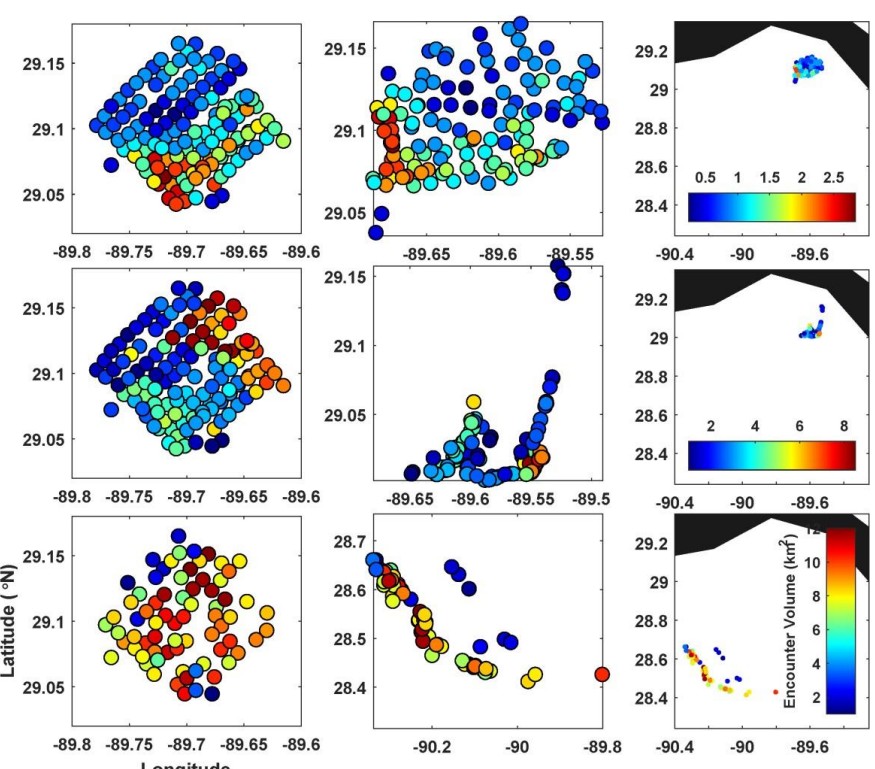

**Figure 6. Real-drifter-based encounter volume at (top) 0.5, (middle) 1, and (bottom) 3 days, mapped to the initial (left) and current (middle and right) positions of drifters.**





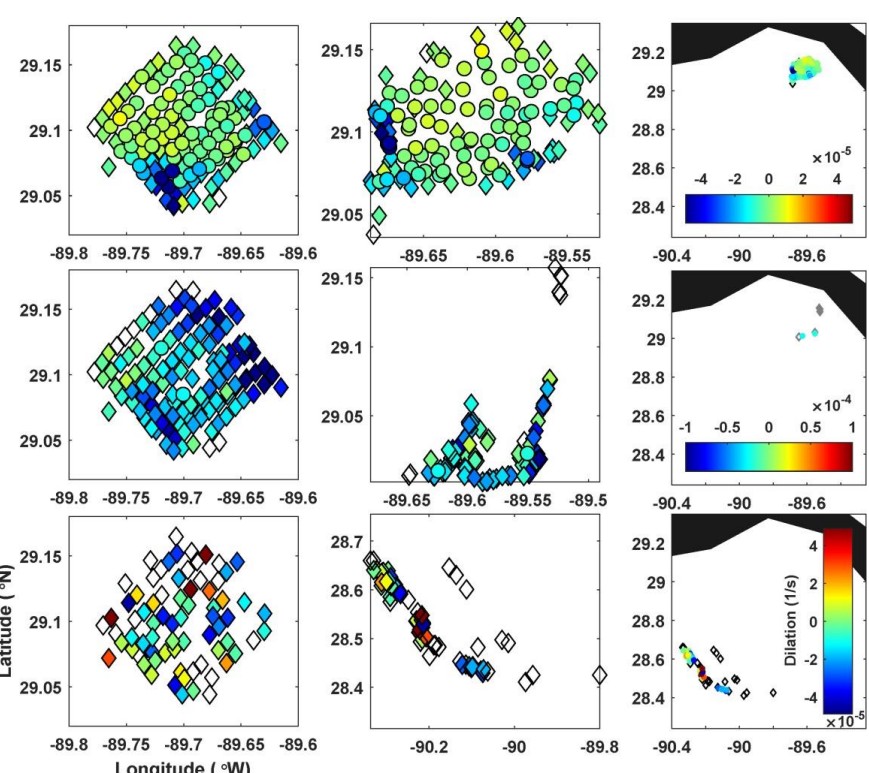

**Figure 7. Real-drifter-based dilation at (top) 0.5, (middle) 1, and (bottom) 3 days, mapped to the initial (left) and current (middle and right) positions of drifters.**



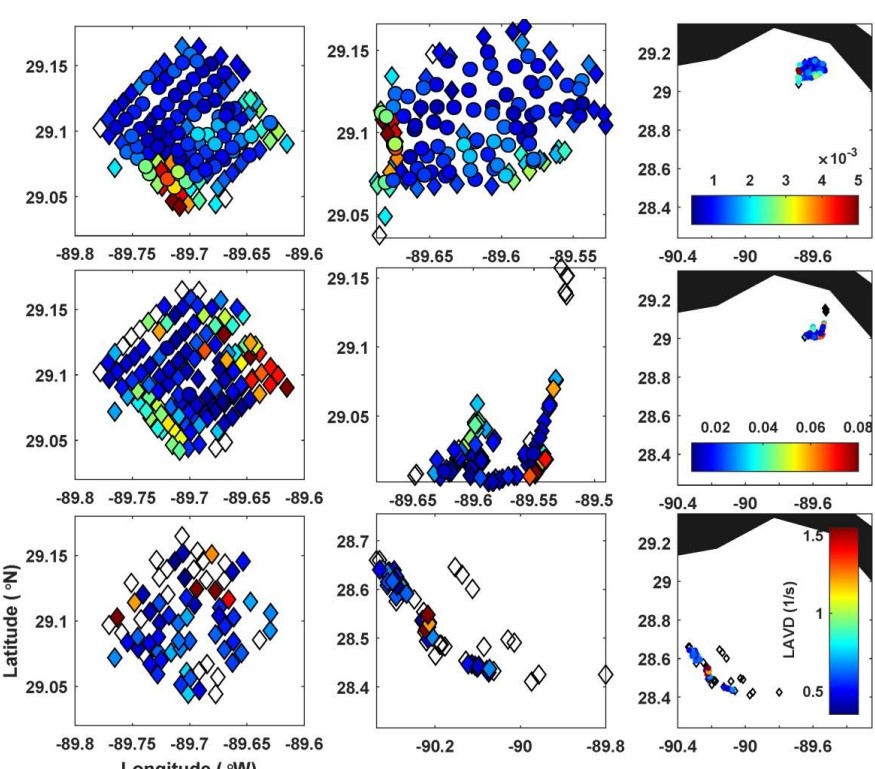


**Figure 8. Real-drifter-based LAVD at (top) 0.5, (middle) 1, and (bottom) 3 days, mapped to the initial (left) and current**
**(middle and right) positions of drifters.**



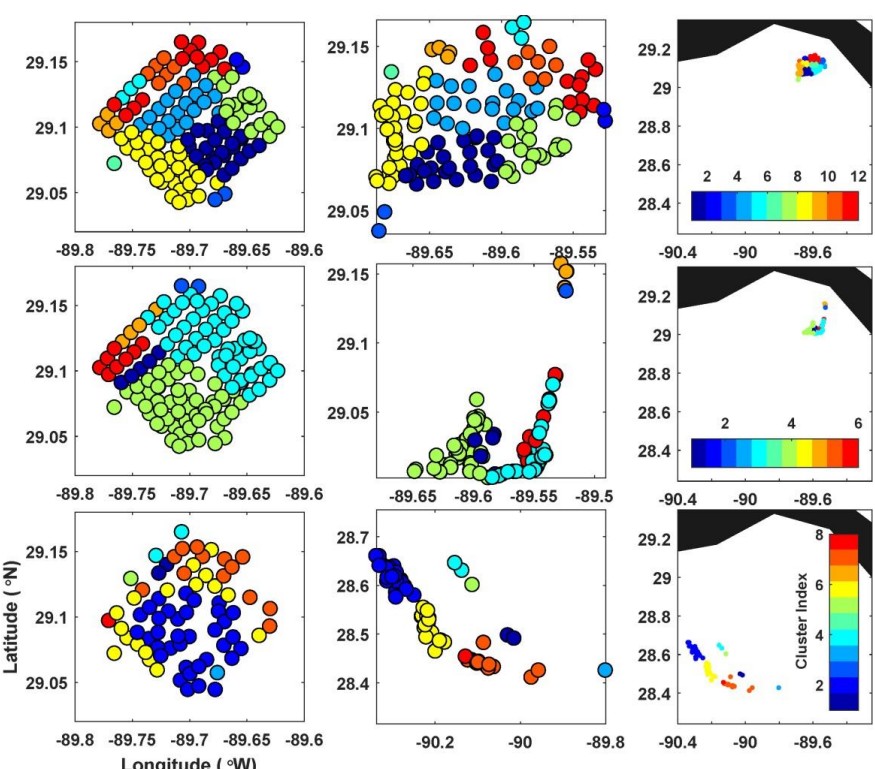


**Figure 9. Real-drifter-based spectral clusters at (top) 0.5, (middle) 1, and (bottom) 3 days, mapped to the initial (left) and current (middle and right) positions of drifters.**







