# Peer review of "Applying dynamical systems techniques to real ocean drifters"

_EGUsphere, 2022_

## Referee Comment (RC1)

**Summary**

This paper makes use of a (comparitively) high resolution set of drifter data to test an array of LCS detection methods that have, in the past, predominantly been applied in idealised numerical experiments or on artificial Lagrangian particles released in observed ocean currents. Application of these methods to detecting LCS in real ocean currents is a long-running motivation for their development, and this work is an important step in that direction.

For the most part, the paper is thorough and well-explained. However, I do have some revisions to suggest, particularly focused on the difficulties of interpreting the LCS methods on such a sparse dataset. A little extra care in linking the results of this study to other datasets would go a long way in strengthening the argument that useful/robust LCS signals have been detected from the SPLASH dataset.

**Comments**

Some of my suggestions are as follows:

**Line 82** Do you mean high costs?

**Line 243** Do the other methods of divergence/vorticity estimation also break down as the aspect ratio increases? Given that filamentation is bound to occur on any set of particles considered, this aspect ratio condition seems to be very detrimental to the discussion of dilation and LAVD.

To improve this discussion, I think the resulting dilation/LAVD from using different methods should be included, or there should be a more detailed explanation as to why these methods were not employed.

If the lack of confidence in the dilation and LAVD are simply natural consequences of applying these measures to a data set where the exact divergence and vorticity are not known , then that is in itself an interesting result, that has wider implications for the potential use of these two measures in future investigations. However, if that is the case, then I believe it should be a more predominant part of the discussion.

**NCOM model comparisons** Given the small number of drifter trajectories (compared to the number of Lagrangian particles that would typically be considered in dynamical systems applications), claims that the results indicate the presence of LCS are quite tentative without additional evidence. The NCOM and simulated 'SPLASH-like' data contained in the supplementary material appears to provide this evidence, so why is there so little reference to this in the main text?

I would suggest that there should at least be some demonstration of the background density, surface velocity fields and 'true' fields from the NCOM data sets in the main text of the paper. These would demonstrate the presence of the fronts, as well as the large scale anticyclonic feature which are argued to be the source of the LCS signals in the SPLASH dataset and the reason the frame-dependent.

---

## Referee Comment (RC2)

**Review of:** *Applying dynamical systems techniques to real ocean drifters,* by Rypina et al.

This manuscript describes a variety of Lagrangian diagnostic fields computed from a dense drifter array deployed in the Gulf of Mexico. These types of fields have long been used to study flow properties and Lagrangian coherent structures (LCS) in simulations, but this is the first time they have been applied to this degree to an observed drifter dataset. As such, this is a welcome contribution to the field. The authors were able to identify a structure reminiscent of a hyperbolic region, as well as areas of strong cumulative convergence and divergence. However, these features were identified from the trajectories alone and only confirmed to a smaller or greater degree by the various diagnostic fields. It remains unclear to me whether any of the calculations added anything to the understanding of the flow. (I should emphasize that I find this a useful report, even if the conclusion is that even this dense drifter deployment is insufficient to extract any use from the diagnostic fields.) The authors' assessments that one field or another was "useful in identifying the dominant LCS" (or not) seemed a bit haphazard. Similarly, the comparisons between a dense and a sparse drifter simulation, presented in the supplemental materials, was rather subjective, and "good agreement" was hard to distinguish from "poor agreement". I believe the paper could benefit if the authors carefully defined what they mean by LCS (it seems to vary for different diagnostics), and clarify the specific contributions of each field, if any. I therefore recommend **major revisions** prior to acceptance of the paper. Detailed comments follow.

Main text

- Line 29: I suggest to name the two unsuccessful methods in the abstract.
- Lines 40 – 43: I find this statement full of jargon and hard to understand for anyone not familiar with Lagrangian ocean analysis.
- Lines 74 – 79: The dataset referenced on line 567 contains 144 drifters released that day in an approximately 12 x 12 grid by 3 boats in just under 3 hours. Maybe a subset of this grid is used here? If so, the authors should specify how and why they subsampled the data. (Regardless, it should definitely not be 7 boats on line 79.) The authors should also specify what date and time they chose as $t_{start}$.
- Line 96: Hadjighasem et al. (2017) also used an observed flow (wind velocity in Jupiter's atmosphere).
- Lines 107 – 109: Could the authors supply a reference for the statement about the spiraling FTLE structures?
- Line 116: Lekien and Ross (2010) is an odd choice for a reference here, since that paper specifically focused on using unstructured meshes. I believe Rypina et al (2021) also used the unstructured mesh method rather than dense regularly spaced orthogonal grids. On a related note, I am surprised the authors do not cite the latter paper as the first application of FTLE to an observational drifter dataset.

- Line 129:  Singular values are by definition always positive.
- Line 132:  Since the rest of the paper refers to 2D flows only, for consistency it might be clearer to stay in 2D here.
- Lines 145 – 146:  Arclength was first proposed by Mendoza & Mancho (2010, doi: 10.1103/PhysRevLett.105.038501) as a Lagrangian descriptor.  Mancho et al. (2013) explored several Lagrangian descriptors, including arclength.
- Lines 169 – 184:  Can the encounter number be appropriately determined from such a limited sample?  After just a short time period, as the drifters separate, the water masses encountered by each drifter are not captured by the other drifters.  It seems that counting only parcels that were initially close to the target particle (i.e., within the deployment field) produces exactly the opposite effect of capturing mixing potential.  Drifters that wandered off on their own are more likely to represent water masses mixing with new waters, no?
  In this context of limited sampling, the authors should discuss what this calculation represents.
- Line 186:  Following the nomenclature of Huntley et al. (2015), this is the dilation *rate*, with units of inverse time.
- Lines 195 – 201:  LAVD was introduced in the context of a flow field that contains coherent vortices as small subsets of the domain.  It is not clear – as the authors note on lines 412 – 414 – how to interpret a vorticity deviation where the entire sample is within a vortex.  Without such an interpretation, is the calculation meaningful?
- Line 203:  This may be a bit nit-picky, but I would consider spectral clustering a data science technique rather than a dynamical systems technique.
- Line 237:  What is the distance condition?
- Line 272:  Could the authors explain why they are using t = 0 as the starting point for all three calculations, instead of considering the intervals [0, 0.5], [0.5, 1], and [1,3]?  The latter approach would truly split the movement into 3 separate stages.
- Lines 275 – 277:  This movie is not included in the supplementary materials (or I couldn't find it).
- Lines 278 – 283:  My impression of the early FTLE field (top left of Fig. 3) is that it does not show any coherent structures, especially if one compares it to the top left field of Fig. S3, where even the SPLASH-like calculation shows clear patterns.
  Line 279 asserts generally smaller values in the south, but some of the highest values occur at the southern end, and larger values in the middle latitudes, but some of the lowest values are found there.  Maybe if FTLE were plotted as a function of latitude only, such a pattern might arise, but I cannot discern it from the presented evidence.
  I am also not convinced that the drifters that are more tightly clustered at time 0.5 are associated with low FTLE (lines 281 – 283), since some of the highest FTLE values also occur in these tight clusters.

- Line 292: Should this be a more compact group in the southern part of the distribution?
- Line 296: It is not necessarily true that the drifters remained close together; they could have separated and then come back together. (Note that some of these data points are colored yellow in the top row.)
- Line 306: It is very hard to distinguish positive from negative FTLE in the plots. [See also my first comment below on the figures.]
- Line 315: While there is generally an increase from south to north, it is hardly monotonic at any of the analysis times across all longitudes.
- Line 330: What kind of "LCS" were the authors looking for that they did not find here? Of all the fields, Fig. 4 looks the most structured to my eye.
- Line 335 – 336: What is meant by a "slightly stronger variability"? To my eye, the variability looks comparable and, if anything, less for CD; but it depends completely on the chosen colors in the plot... How is the variability in two quantities that have different units like these compared? The standard deviation as a percentage of the mean?
- Lines 338 – 339: Why would one expect CD to be indicative of convergence? It seems like CD is not the right diagnostic for the LCS the authors were hoping to identify here, just based on its definition.
- Line 343: The variability at earlier times looks fairly small. But again, how should this be assessed? Is a range from 0.5 to 3 big or small in this context? Maybe it would help to determine the possible minimum and maximum values achievable over the different time intervals.
- Lines 353 – 354: It is counter-intuitive that filamenting water parcels would be encountering fewer neighbors than parcels remaining more compact. This is solely a function of undersampling of the encountered neighbors. (See comment above for lines 169 – 184.)
- Line 476: It would be helpful to explicitly identify/summarize the dominant LCS being referenced here, since most of the plots did not exhibit much obvious coherence.
- Line 491: Should this be the spectral clusters other than green?
- Lines 535 – 537: Another possibility is that LAVD is the wrong tool for identifying a coherent eddy core from a sample of significant mean vorticity.
- Line 550: I am not a big fan of calling a numerically derived model field the "true" model field, since it is also subject to errors, albeit smaller than in the "SPLASH-like" calculation. Maybe a better choice would be to call it a dense simulation.
- Lines 552 – 556: I think this is overstating the case a bit. The two model simulations showed good agreement of the coherent features in some cases, but not in others (e.g., late time FTLE, early and medium time $V_{en}$, early and late time clusters).
  I would also argue that, especially in the FTLE field, the model calculations show much greater coherence than those from the observations. I'm not sure what the similarities

in the geometries and types of features are that are being referenced on line 554. The patterns from the model seem quite different from those in the observations (e.g., mostly negative $D$ in the early observations vs. mostly positive $D$ in the early simulations; model fields are generally more coherent, especially at the late time). Lastly, I think the model experiment can show reliability but not robustness.

- It may be worth commenting in the Summary & Discussion on the differences between the frame-dependent and the frame-independent diagnostic fields and how they should be used differently.

References

- The references should be alphabetized.
- Filippi et al. (2021) should be given an 'a' and 'b' to differentiate the two publications.
- Froyland and Padberg-Gehle (2015) is not referenced in the manuscript.
- Essink et al. (2021) is missing from the references.
- It seems that the references for the supplemental materials are included here. If so, Beron-Vera et al. (2015) is also missing.

Figures

- Across all figures (main and supplemental), the choice of colormaps is not ideal. The authors try to compare fields between different rows when they have different scales on the colormaps. For some quantities (e.g. FTLE), the colormap is strangely designed to highlight gradients in very specific narrow bands only. Some of the quantities (FTLE and dilation rate), the distinction of positive and negative values is important, but hard to do with the given colormaps. Thus, I recommend modifying the colormaps such that they vary continuously, except that for FTLE and dilation rate there is a clear break at 0. I also recommend using uniform ranges on the colorbars for quantities that are scaled by or independent of the time interval (FTLE, correlation dimension, dilation rate, LAVD). For the clusters, the number of different colors in the colormap should equal the number of clusters. It would also help if they were chosen to be more easily distinguishable, as currently some of the shades of blue and some of the shades of red are hard to tell apart.
- Aside from providing the colorbars, the right column of the figures is not needed, since the drifter positions are placed in geographic context in Figs. 1 and S2.
- Figs. 7 and 8: It would help to have a reminder in the caption here what the different symbols mean and why some diamonds are left white.

Supplementary materials

- Lines 21 – 24: Doesn't the topography also play a role here, in addition to the density gradients? And wind?

- Line 51: Why were different time intervals chosen in the model than for the observations? Line 47 suggests possibly using 1.5 days instead of 1 day, but why 2 days, and then 4 days?

- Lines 63 – 64: I don't see the similarities in the patterns of high and low values. At early times: observations – scattering of a few high values near the corners of the deployment; model – coherent swath of high values in the center of the domain. At late times: observations – mostly high values throughout the domain, with a region of lower values in the lower half away from the edges; model – mostly low values, with a swath of very high values along the SE edge.
The magnitude of the values is also not comparable, if one can use the ranges on the colorbars as a guide, especially for negative FTLE values.

- Line 83: It is not clear in what ways the agreement was favorable. E.g., one could not identify the coherent structures at the middle time from the SPLASH-like simulation alone, even if they are weakly reflected in it.

- Line 96: I am not clear what standards of agreement are being used here. The structures in the dense simulation (S-shaped and longitudinal ridges) are not identifiable in the SPLASH-like simulation. Magnitudes are impossible to compare, since different colorbar ranges are used. Maybe a scatter plot of values at the positions of the SPLASH-like data from the two simulations would make that clearer. (Such scatter plots could replace the right columns of Figs. S3 – S8 to directly assess the reliability of the sparse calculations for all the Lagrangian diagnostics.)

- Line 109: The SPLASH-like fields shows patches of increased $V_{en}$ similar to those deemed meaningful in the observations. Why are they considered negligible here?

- Line 125: Are they confined to a smaller area, or are they hidden behind the large markers?

- Line 156: I would not say that the SPLASH-like simulation successfully mapped out the dominant features at 4 days, although it did seem to show a hint of the region of lower LAVD values in the middle.

- Line 168: Fig. S9 shows only 4 clusters for the second version at intermediate times.

- Line 191: Is it meaningful to compare the number of clusters found in the two simulations over different domains? Maybe it would be better to apply the spectral clustering in the dense simulation only to the subdomain also sampled by the SPLASH-like simulation.

- Line 203: I disagree that the clusters look similar in the figures. The only similarity I find is the split around 29N.

- Line 213: I suggest citing the filtering method developed for these kinds of trajectories by Yaremchuk & Coelho (2015, doi: 10.1109/JOE.2014.2353472).

- To draw comparisons with the observations (e.g. "similar magnitudes"), it would be helpful to use the same colormaps and ranges in the supplemental figures as are used in the main manuscript for the corresponding diagnostics.
- Fig. S4:  I don't understand why the values shown along the longitudinal filament in the south at the late time don't agree between the two simulations.  Coarse sampling should have no impact on pathlength.  Is this a plotting artifact?

*Language*

Below are a few suggestions to fix grammar and spelling issues.  Overall, this is one of the most readable manuscripts I have reviewed lately; so these are just minor things.

- Line 68: "and then use observations of their trajectories"
- Line 91: "parallels were drawn between…"
- Line 123: "coordinate system
- Line 150: "frame-independent"
- Line 210: "distance between the $i^{th}$ and $j^{th}$ trajectories"
- Line 216: "the cluster sizes"
- Line 225: "The LLS methods"
- Lines 226 and 229: The vectors for U and A should be transposed.  (They need to be n x 1 and 3 x 1, respectively.)
- Line 240, 247: "LLS"
- Line 262: "Some clustering temporarily occurs" (no "of")
- Lines 338 – 339: "neither" and "nor" should be replaced with "either" and "or"
- Line 463: Strike "the" from "the NOAA's Global Drifter Program"
- Line 474: Strike "respectively" (respective to what?)
- Line 526: "The SPLASH experiment was…"
- Line 542: "In the SPLASH experiment"
- Line 710: "experiment sight"
- Line 731: "(bottom) 3 days"
- Supp, Line 8: "the site of the SPLASH experiment"
- Supp, Line 29: "as is evident"
- Supp, Line 91: "Fig. S3"
- Supp, Line 117: "blue values"
- Supp, Line 121: "a bit of red"
- Supp, Line 206: I suggest spelling out STD.
- Supp, Line 208: "$V_{en}$" instead of "$N_{en}$"

---

## Author Comment (AC1)

**Reviewer 1**

Summary

This paper makes use of a (comparatively) high resolution set of drifter data to test an array of LCS detection methods that have, in the past, predominantly been applied in idealized numerical experiments or on artificial Lagrangian particles released in observed ocean currents. Application of these methods to detecting LCS in real ocean currents is a long-running motivation for their development, and this work is an important step in that direction.

For the most part, the paper is thorough and well-explained. However, I do have some revisions to suggest, particularly focused on the difficulties of interpreting the LCS methods on such a sparse dataset. A little extra care in linking the results of this study to other datasets would go a long way in strengthening the argument that useful/robust LCS signals have been detected from the SPLASH dataset.

We thank the reviewer for the overall positive feedback on our work and for thoughtful comments and suggestions. Our point-by-point answers to the reviewer's comments and questions are below.

Comments

Some of my suggestions are as follows:
1) Line 82: Do you mean high costs? Yes, high costs, thank you for finding the typo.
2) Line 243: Do the other methods of divergence/vorticity estimation also break down as the aspect ratio increases? Given that filamentation is bound to occur on any set of particles considered, this aspect ratio condition seems to be very detrimental to the discussion of dilation and LAVD. To improve this discussion, I think the resulting dilation/LAVD from using different methods should be included, or there should be a more detailed explanation as to why these methods were not employed.

Yes, other methods also break down as the aspect ratio of the drifter polygon increases. In the revision we have extended the methods section to include more detail about the different methods for estimating divergence and vorticity from drifters, provided an additional reference where detailed comparison between different estimates has been done, and added a paragraph about this to the discussion section.

The new text in the methods section now reads:

"Note that methods other than LLS can also be used to compute divergence and vorticity.

Specifically, divergence can be estimated as a rate of change of the area spanned by the drifter

polygon, and both divergence and vorticity can be estimated using Green's theorem as,

respectively, the circulation around and total flux through the drifter polygon. Rypina et al.

(2021) compared all three techniques in detail using both real and simulated drifters deployed in the Alboran Sea at similar inter-drifter distances as the SPLASH drifters, and observed good correspondence between all three techniques for clusters of 6 drifters, as long as the drifters stayed within a few km of each other and the aspect ratio was reasonably small (≤5). For larger aspect ratios all methods started to deteriorate. Essink et al. (2022) also investigated the optimal way of computing velocity gradients, divergence, and vorticity from drifters. By quantifying the uncertainty in the velocity gradient calculation for different methods and different drifter configurations in a high-resolution submesoscale-resolving ocean circulation model, they concluded that the LLS was the most robust among the three methods, and that the accuracy of the LLS estimates grew linearly with the increasing number of drifters, and decreased logarithmically with the increasing aspect ratio of the drifter polygon (i.e., LLS works best for tight equidistant polygons with many drifters). Based on their analysis, they favored LLS over the area-rate-of-change and Green's theorem methods as their preferred method, and proposed 6 drifters with a polygon length scale of about 10 km and an aspect ratio of less than 10 as an optimal parameter range for reliable estimation of velocity gradients. They then successfully used LLS with these parameter criteria for estimating divergence and vorticity from the drifters in the Bay of Bengal.

Guided by recommendations of Rypina et al. (2022) and Essink et al. (2021), in this paper we will rely on the LLS method for estimating velocity gradients, and will refer to the LLS estimates of divergence and vorticity as trustworthy (and mark them by colored circles) if there are $\geq 6$ drifters within a $3 -$km radius, the center of mass of the drifter distribution is located within the polygon, and the polygon aspect ratio is $\leq 6$. If only the aspect ratio condition is not

satisfied (but the number of drifters, the distance, and the center of mass conditions are), we

will still compute LLS estimates but we will refer to them as less trustworthy (and mark them by

colored diamonds). In all other cases, we do not produce estimates of divergence and vorticity."

We also follow up on this in the discussion section as described in our answer to next comment.

3) If the lack of confidence in the dilation and LAVD are simply natural consequences of applying these measures to a data set where the exact divergence and vorticity are not known, then that is in itself an interesting result, that has wider implications for the potential use of these two measures in future investigations. However, if that is the case, then I believe it should be a more predominant part of the discussion.

We agree that this point deserves more attention. In the revision, we have added the following paragraph to the discussion:

"One challenge with dilation and $LAVD$ is the loss of accuracy at longer times, when the drifters

form elongated polygons. The deterioration of the velocity gradients estimates (that are

required for estimating dilation and $LAVD$) with the increasing aspect ratio of the drifter

polygon is intuitively clear. As the polygon elongates, the information about the velocity

gradient in the perpendicular direction diminishes and is lost when the polygon approaches a

one dimensional line. This is true for all methods of estimation, not just for LLS, and presents a

fundamental challenge for estimating dilation and $LAVD$ from drifters, which tend to naturally

form elongated filaments in oceanic flows."

4) NCOM model comparisons: Given the small number of drifter trajectories (compared to the number of Lagrangian particles that would typically be considered in dynamical systems applications), claims that the results indicate the presence of LCS are quite tentative without additional evidence. The NCOM and simulated `SPLASH-like' data contained in the supplementary material appears to provide this evidence, so why is there so little reference to this in the main text? I would suggest that there should at least be some demonstration of the background density, surface velocity fields and `true' fields from the NCOM data sets in the main text of the paper. These would demonstrate the

presence of the fronts, as well as the large scale anticyclonic feature which are argued to be the source of the LCS signals in the SPLASH dataset and the reason the frame-dependent.

In the revision, we have extended the text in the discussion section describing model simulations, but decided to not include any more figures into the paper. Our main reasoning for confining most model-based results to Supplementary Material was keeping our paper as short and clear as possible. With 7 different methods and 9 multi-panel figures, the paper is already quite long. As pointed out by the reviewer, the main novelty of our work is that various LCS identifiers were applied to real drifters, and we didn't want to distract the reader by shifting the emphasis too much towards simulated drifters. Supplementary Material will be published together with the paper, so all interested readers should be able to easily access it.

**Reviewer 2**

This manuscript describes a variety of Lagrangian diagnostic fields computed from a dense drifter array deployed in the Gulf of Mexico. These types of fields have long been used to study flow properties and Lagrangian coherent structures (LCS) in simulations, but this is the first time they have been applied to this degree to an observed drifter dataset. As such, this is a welcome contribution to the field.

We thank the reviewer for the positive feedback on our paper and for the thoughtful comments and suggestions.

The authors were able to identify a structure reminiscent of a hyperbolic region, as well as areas of strong cumulative convergence and divergence. However, these features were identified from the trajectories alone and only confirmed to a smaller or greater degree by the various diagnostic fields. It remains unclear to me whether any of the calculations added anything to the understanding of the flow. (I should emphasize that I find this a useful report, even if the conclusion is that even this dense drifter deployment is insufficient to extract any use from the diagnostic fields.)

One challenge that complicates the understanding of the flow from this type of analysis is that the features that are found (for example, a hyperbolic region) say something about local kinematic features of the flow but do not allow one to say much about how the flow looks like on broader spatial scales, or over time scales longer than just a few days. The rapid filamentation experienced by the drifters prevents mapping out the structures at later times and over regions other than the original deployment domain. This might be one of the important things that we have learned about the flow, and about sampling through massive drifter releases. If we were able to saturate the whole Gulf of Mexico with drifters and keep it saturated for a long time, as in a numerical model, we probably would have been able to learn a lot more about the kinematics, and perhaps certain dynamical aspects of the flow. We have added a comment about this to the conclusions section.

The authors' assessments that one field or another was "useful in identifying the dominant LCS" (or not) seemed a bit haphazard. Similarly, the comparisons between a dense and a sparse drifter simulation, presented in the supplemental materials, was rather subjective, and "good agreement" was hard to distinguish from "poor agreement". I believe the paper could benefit if the authors carefully defined what they mean by LCS (it seems to vary for different diagnostics), and clarify the specific contributions of each field, if any. I therefore recommend **major revisions** prior to acceptance of the paper. Detailed comments follow.

As suggested by the reviewer, in the revised methods section we have more clearly described what types of LCS are highlighted by the different Lagrangian diagnostic fields (lines 105-110, 149-152, 166-175, 181-185, 214-217, 223-229, and 235-241). We also tried to be more cautious and less subjective in our assessment of the good and poor agreement. In many places throughout the manuscript, we now use language suggested by the reviewer when making our comparisons.

Our point-by-point answers to the reviewer's comments are below.

Main text

1) • Line 29: I suggest to name the two unsuccessful methods in the abstract. - implemented

2) • Lines 40 – 43: I find this statement full of jargon and hard to understand for anyone not familiar with Lagrangian ocean analysis.

We have eliminated jargon and the new statement now reads:

"Over the last several decades, techniques from the dynamical systems theory have been extensively used to study transport in oceanic flows. However, these methods have been typically applied to numerically simulated trajectories of water parcels. This paper applies different dynamical systems techniques to real trajectories of 144 ocean surface drifters from the massive drifter release in the Gulf of Mexico. To our knowledge, this is the first comprehensive comparison of the performance of several different dynamical systems techniques with applications to real ocean drifters."

3) • Lines 74 – 79: The dataset referenced on line 567 contains 144 drifters released that day in an approximately 12 x 12 grid by 3 boats in just under 3 hours. Maybe a subset of this grid is used here? If so, the authors should specify how and why they subsampled the data. (Regardless, it should definitely not be 7 boats on line 79.) The authors should also specify what date and time they chose as $t_{start}$.

Thank you for finding the typos, the dataset indeed contains 144 drifters released in an approximately 12 x 12 grid by 3 boats in just under 3 hours. We used all 144 drifters, and the start time $t_{start}$ for our analysis corresponds to the time when the last drifter was released (the drifter positions at $t_{start}$ are shown in Fig. 1). We have clarified this in the revision.

4) • Line 96: Hadjighasem et al. (2017) also used an observed flow (wind velocity in Jupiter's atmosphere).

Yes, added "…, observed, and…" to the sentence.

5) • Lines 107 – 109: Could the authors supply a reference for the statement about the spiraling FTLE structures?

We have often observed FTLE spiraling within ocean eddies in our prior work. The most relevant reference that we are aware of is Munk et al., 2000 [Walter Munk, Laurance Armi, Kenneth Fischer and F. Zachariasen, 2000. Spirals on the Sea. Proceedings of the Royal Society

A. Https://doi.org/10.1098/rspa.2000.0560]. However, because FTLEs were not specifically discussed in that paper, we have removed the sentence in question from the paper.

6) • Line 116: Lekien and Ross (2010) is an odd choice for a reference here, since that paper specifically focused on using unstructured meshes. I believe Rypina et al (2021) also used the unstructured mesh method rather than dense regularly spaced orthogonal grids. On a related note, I am surprised the authors do not cite the latter paper as the first application of FTLE to an observational drifter dataset.

In the revision, we now cite Haller (2001) instead of Lekien and Ross (2010) as the most relevant reference for computing FTLEs on regular meshes. [Haller G. 2001. Distinguished material surfaces and coherent structures in three-dimensional fluid flows. *Physica D* 149:248–77]

We also added the following sentence to the paper referencing Rypina et al. (2021):

"A modification for unstructured meshes was described in Lekien and Ross (2010). Rypina et al. (2021) recently used the unstructured grid method to compute FTLEs from a cluster of 6 real drifters in the Alboran Sea."

7) • Line 129: Singular values are by definition always positive.

Removed "positive" from the sentence.

8) • Line 132: Since the rest of the paper refers to 2D flows only, for consistency it might be clearer to stay in 2D here.

Agree. We used the 2D formulation in the revision.

9) • Lines 145 – 146: Arclength was first proposed by Mendoza & Mancho (2010, doi: 10.1103/PhysRevLett.105.038501) as a Lagrangian descriptor. Mancho et al. (2013) explored several Lagrangian descriptors, including arclength.

Thank you for the correction. We now cite Mendoza and Mancho (2010) in the revision.

10) • Lines 169 – 184: Can the encounter number be appropriately determined from such a limited sample? After just a short time period, as the drifters separate, the water masses encountered by each drifter are not captured by the other drifters. It seems that counting only parcels that were initially close to the target particle (i.e., within the deployment field) produces exactly the opposite effect of capturing mixing potential. Drifters that wandered off on their own are more likely to represent water masses mixing with new waters, no? In this context of limited sampling, the authors should discuss what this calculation represents.

We thank the reviewer for this thoughtful comment. We agree that in the context of limited trajectories deployed in a small part of the domain, the interpretation of the encounter volume as a measure of mixing potential is not straightforward and differs from that in the case where drifters are seeded over the entire domain. We have added the following paragraph to address this issue:

"Note that the interpretation of the encounter volume in the context of limited trajectories deployed in a small part of a flow domain, such as our SPLASH drifters, differs from the case where drifters are seeded over the entire domain. Only for a domain-wide deployment, encounter volume is representative of the mixing potential of the flow. For a small deployment, encounter volume merely measures the amount of encounters within the dataset. This undersampling issue leads to important consequences in both hyperbolic and elliptic regions. While for a domain-wide deployment a lot of encounters occur in hyperbolic regions (as discussed above), these are also the exact same regions where initially-nearby trajectories separate rapidly from each other, yielding low encounter values in the case of a small deployment. Similarly, whereas coherent eddy cores produce fewer encounters than hyperbolic regions for a domain-wide drifter release, these regions trap drifters allowing them to encounter many of their neighbors deployed within the same eddy, which produces large values in the case of small deployment. Thus, encounter volume might be a poor measure of the mixing potential of a flow in the case of a small deployment (but because this metric is still sensitive to differences between hyperbolic/elliptic behaviors even for a small deployment, it might still be able to highlight regions with different transport characteristics, so we go ahead and apply it to SPLASH drifters in the next section)."

11)• Line 186: Following the nomenclature of Huntley et al. (2015), this is the dilation *rate*, with units of inverse time.

Corrected, thank you.

12)• Lines 195 – 201: LAVD was introduced in the context of a flow field that contains coherent vortices as small subsets of the domain. It is not clear – as the authors note on lines 412 – 414 – how to interpret a vorticity deviation where the entire sample is within a vortex. Without such an interpretation, is the calculation meaningful?

We agree with this comment and have added a sentence about this to the section on LAVD:

"Note that *LAVD* would only be able to identify those rotationally-coherent Lagrangian eddies that are smaller than, and lay entirely within, the domain seeded with drifters."

For SPLASH drifters, LAVD did not show any closed convex contours surrounding local isolated maxima. In this sense, the calculation was meaningful and suggested that there were no rotationally-coherent Lagrangian eddies within the domain seeded with drifters. We think that during the initial stage LAVD did not identify the anticyclonic eddy because the deployment domain was too small and was located entirely within the vortex. We added the following text to clarify this:

"Note that our high-*LAVD* regions differed from the classical examples of rotationally-coherent Lagrangian eddies. Our regions were not circular, did not have a single maximum, and were too noisy to identify the outermost convex contour level, which marks the outer edge of the coherent rotational eddies in the standard application of the *LAVD* technique. Thus we cannot call these high-*LAVD* features rotationally-coherent Lagrangian eddies. It is interesting that even though trajectories exhibited clear anticyclonic rotation during the first 12 hours, *LAVD* did not identify this anticyclonic eddy. We think this might be because the SPLASH release domain was too small and was located entirely within this vortex structure."

13)• Line 203: This may be a bit nit-picky, but I would consider spectral clustering a data science technique rather than a dynamical systems technique.

We have added a sentence about this: "This was originally a data science technique that was adopted by the dynamical systems community."

14)• Line 237: What is the distance condition?

The distance condition is that 6 or more drifters are located within a circle with a 3-km radius. We have clarified this.

15)• Line 272: Could the authors explain why they are using t = 0 as the starting point for all three calculations, instead of considering the intervals [0, 0.5], [0.5, 1], and [1,3]? The latter approach would truly split the movement into 3 separate stages.

Since all 7 identifiers map out LCS at the start time of trajectory, using t=0 as the start time allowed to make the best use of the nearly-regular deployment pattern. Using other time intervals, [0.5, 1] and [1,3], results in identifiers mapped to the location of trajectories at t=0.5 and t=1, respectively, when drifters already stretched into a highly elongated irregular pattern.

We have added a paragraph clarifying this point:

"Having split the drifter movement into 3 stages, we next apply our Lagrangian methods to trajectory segments from $t_{start} = 0$ days until $t_{end} = 0.5$, 1, and 3 days, respectively (top, middle, and bottom row of panels in Figs. 3-9). The resulting fields highlight the dominant LCSs that existed at the time of the drifter deployment (i.e., at $t_{start} = 0$) and that governed the movement of drifters during the subsequent 0.5, 1, and 3 days, respectively. (A movie of the FTLEs at $t_{start} = 0$ computed with progressively increasing $t_{end}$ is included in the Supplementary Material.) Since all 7 identifiers map out LCS at the start time of trajectory, using $t_{start} = 0$ allowed making the best use of the nearly-regular deployment pattern. Fields computed for other time intervals, for example, [0.5, 1] day or [1, 3] days, would need to be mapped to the location of trajectories at 0.5 days and 1 day, respectively, when drifters already stretched into highly elongated filaments, thus losing the advantage of the regular deployment grid."

16)• Lines 275 – 277: This movie is not included in the supplementary materials (or I couldn't find it).

We thank the reviewer for letting us know that the movie was not available. We will make sure it is accessible in the revised manuscript.

17)• Lines 278 – 283: My impression of the early FTLE field (top left of Fig. 3) is that it does not show any coherent structures, especially if one compares it to the top left field of Fig. S3, where even the SPLASH-like calculation shows clear patterns. Line 279 asserts generally smaller values in the south, but some of the highest values occur at the southern end, and larger values in the middle latitudes, but some of the lowest values are found there. Maybe if FTLE were plotted as a function of latitude

only, such a pattern might arise, but I cannot discern it from the presented evidence. I am also not convinced that the drifters that are more tightly clustered at time 0.5 are associated with low FTLE (lines 281 – 283), since some of the highest FTLE values also occur in these tight clusters.

We agree that the early FTLE field does not show any clear coherent structures, and the revised text now clearly stated this:

"During the initial stage of motion ($t_{start} = 0$ days and $t_{end} = 0.5$ days), the FTLE field did not show any clear coherent structures, neither hyperbolic (maximizing ridges) nor elliptic (isolated regions with significantly lower FTLEs)."

18) • Line 292: Should this be a more compact group in the southern part of the distribution? – yes, thank you

19) • Line 296: It is not necessarily true that the drifters remained close together; they could have separated and then come back together. (Note that some of these data points are colored yellow in the top row.)

Correct, we have changed the sentence to read:

"This cluster contained trajectories that either remained together, or separated and then came back together (since some of these data points are marked by yellow in the top row)."

20) • Line 306: It is very hard to distinguish positive from negative FTLE in the plots. [See also my first comment below on the figures.]

We have changed the colorbar to have a more clearly visible change (blue/black) at 0.

21) • Line 315: While there is generally an increase from south to north, it is hardly monotonic at any of the analysis times across all longitudes.

We removed "monotonic."

22) • Line 330: What kind of "LCS" were the authors looking for that they did not find here? Of all the fields, Fig. 4 looks the most structured to my eye.

We were looking for hyperbolic LCS (i.e., suspect manifold during the 2nd stage of motion), which would show up as level sets of $L$ with the highest gradient in the perpendicular direction, and for elliptic LCS (i.e., suspect coherent eddies during the 1st and 3rd stages),

which would correspond to flatly low $L$ with a high gradient towards much higher values at the periphery. We did not find any. The revised text now reads:

"Specifically, we tried looking for hyperbolic LCS, which would show up as level sets of $L$ with the highest gradient in the perpendicular direction, and for slow-moving elliptic regular regions which should be characterized by a uniformly low $L$ with a high gradient toward large $L$ at the periphery, but we did not find any."

23)• Line 335 – 336: What is meant by a "slightly stronger variability"? To my eye, the variability looks comparable and, if anything, less for CD; but it depends completely on the chosen colors in the plot... How is the variability in two quantities that have different units like these compared? The standard deviation as a percentage of the mean?

We agree that variability is not a good measure of the performance of the method. For example, squaring (or square-rooting) the values would change the magnitude of variability but would not provide any new information. We have removed the sentence in question to avoid confusion.

24)• Lines 338 – 339: Why would one expect CD to be indicative of convergence? It seems like CD is not the right diagnostic for the LCS the authors were hoping to identify here, just based on its definition.

Although originally designed to identify hyperbolic LCS (level sets with highest gradient in perpendicular direction), for flows in the state of chaotic advection, CD could also be used to highlight slowly-moving coherent eddy-like features (regular islands), embedded into vigorously-stirring regions (chaotic sea). Islands would have less complex trajectories with lower CD than chaotic trajectories within the chaotic sea. Similarly, although CD was not designed to identify convergence, trajectories converging rapidly into a nearly-stationary convergence zone would have smaller CD (i.e., less complex) than those free to wonder over the entire domain. We have added a paragraph about the ability of CD to highlight regular islands and converging zones to the description of the methods:

"Note also that for flows in the state of chaotic advection, $CD$ (and $L$) could also be used to highlight slowly-moving coherent eddy-like features (regular islands), embedded into vigorously-stirring regions (chaotic sea). Islands would have less complex trajectories with lower $C$ than trajectories within the chaotic sea. Similarly, although $CD$ was not designed to identify convergence, trajectories converging rapidly into a nearly-stationary convergence zone would have smaller $CD$ than those free to wonder over the entire domain."

For SPLASH dataset, however, we found no indication of regular islands or convergence zones in the CD fields.

25) • Line 343: The variability at earlier times looks fairly small. But again, how should this be assessed? Is a range from 0.5 to 3 big or small in this context? Maybe it would help to determine the possible minimum and maximum values achievable over the different time intervals.

Again, we agree that variability of the field is not a good measure of the performance of the method. We have removed the mention of variability.

26) • Lines 353 – 354: It is counter-intuitive that filamenting water parcels would be encountering fewer neighbors than parcels remaining more compact. This is solely a function of undersampling of the encountered neighbors. (See comment above for lines 169 – 184.)

We agree. In the revision, we added clarification about the application of encounter volume to limited datasets. The new text can be found both in the methods section (see our response to comment (10)) and in the results section:

"The elongated along-shore filament seen at $1$ day contained smallest $V_{en}$ since trajectories in the filament separated rapidly from their nearby neighbors and thus did not encounter many SPLASH trajectories. This is likely a consequence of undersampling in hyperbolic regions (note that the same region was marked by largest FTLEs indicative of hyperbolic behavior). Since SPLASH drifters were only seeded over a small O($10\ km^2$) domain, the resulting $V_{en}$ characterizes encounters within this limited dataset, rather that with all trajectories in the entire domain, leading to smallest $V_{en}$ in this hyperbolic region instead of largest $V_{en}$, as would likely have been the case for a domain-wide trajectory deployment."

And then later in the same section:

"$V_{en}$ is also more susceptible to undersampling issues than FTLEs, since the number of encounters within a limited dataset is not necessarily representative of that with trajectories seeded over the entire domain. For SPLASH drifters, undersampling led to smallest $V_{en}$ along the northwestern edge of the release domain during the 2${}^{nd}$ stage of motion, where large FTLEs indicated the presence of a stable manifold of a hyperbolic trajectory that was responsible for the formation of an elongated along-shore filament at 1 day.

Overall, despite some challenges associated with undersampling, $V_{en}$ proved to be an interesting frame-independent diagnostic that was sensitive to both enhanced clustering, hyperbolic behavior, and flow convergence, and was complementary to FTLEs."

27) • Line 476: It would be helpful to explicitly identify/summarize the dominant LCS being referenced here, since most of the plots did not exhibit much obvious coherence.

We have added a description of observed LCS to the sentence in question. LCS are discussed in more detail in the next paragraph.

"… dominant LCS (such as the hyperbolic-type LCS responsible for the formation of the along-shore filament at 1 day, the convergence-type LCS attracting drifters into the southeastern corner at 1 day, and the elliptic-type LCS forming during the 3$^{rd}$ stage of off-shore motion)…"

28) • Line 491: Should this be the spectral clusters other than green? Corrected.

29) • Lines 535 – 537: Another possibility is that LAVD is the wrong tool for identifying a coherent eddy core from a sample of significant mean vorticity.

We agree that LAVD might be a wrong tool for identifying an eddy from a trajectory set located entirely within an eddy. We have added this possibility to the sentence in question.

30) • Line 550: I am not a big fan of calling a numerically derived model field the "true" model field, since it is also subject to errors, albeit smaller than in the "SPLASH-like" calculation. Maybe a better choice would be to call it a dense simulation.

We have changed "true" to "dense-grid simulations."

• Lines 552 – 556: I think this is overstating the case a bit. The two model simulations showed good agreement of the coherent features in some cases, but not in others (e.g., late time FTLE, early and medium time $V_{en}$, early and late time clusters). I would also argue that, especially in the FTLE field, the model calculations show much greater coherence than those from the observations. I'm not sure what the similarities in the geometries and types of features are that are being referenced on line 554. The patterns from the model seem quite different from those in the observations (e.g., mostly negative $D$ in the early observations vs. mostly positive $D$ in the early simulations; model fields are generally more coherent, especially at the late time).

We have incorporated reviewer's suggestions and extended the last paragraph to read:

"Comparison between SPLASH-like and dense-grid model simulations showed reasonably good agreement for many, although not all, metrics and times, suggesting that many, although not all, SPLASH fields were reliable (see Supplementary Material). Specifically, SPLASH-like FTLEs were most reliable at shorter times and still meaningful at longer times in regions with strong hyperbolic-type LCS located far enough away from each other to be resolved by the deployment grid. $L$ and $CD$ were reliable at all times, but since they did not identify any hyperbolic, elliptic, or convergence-type LCS for SPLASH drifters, they were perhaps least useful among the 7 methods. In contrast to FTLEs, $V_{en}$ was not reliable at short times but improved its reliability at longer times. $D$ and $LAVD$ worked well at short times when drifters were still relatively close and didn't form elongated filaments, but deteriorated at longer times due to the rapid filamentation of the drifter distribution. Finally, SPLASH-like and dense-grid SC both identified large numbers of clusters within the SPLASH domain at short times and fewer clusters at longer times. At short time, the clusters were different between the SPLASH-like and dense-grid simulations; at longer times, there was a number of similarities in the identified clusters (longitudinal split along same longitude at intermediate time, and assignment of most of the domain to 1 cluster at later time), but the details of the cluster configurations were different, especially near the edges of the domain.

Comparing observations to simulations, Lagrangian metrics were of similar magnitude for the real and simulated SPLASH drifter. The actual range of values in simulations and observations matched for $FTLEs$ and $D$, as well as $L/CD$ at 0.5 days and 3 days, and $V_{en}$ at 0.5 days. $LAVD$ was 2 to 3 times larger in observations, $V_{en}$ was 2 to 3 times larger in simulations at the intermediate and late stages, and $L/CD$ were slightly larger in simulations at the intermediate stage (note, however, that we used 1 day/2 days as a characteristic time for the intermediate stage

in observations/simulations). Hyperbolic- and convergence-type LCS were present in both observations and simulations, and no clear elliptic-type LCS were seen in either model or observations. The model fields were generally significantly less noisy, exhibited a larger degree of coherence, and at early times had more positive dilation, compared to mostly near-zero and negative in observations. Detailed comparisons can be found in Supplementary Material."

Lastly, I think the model experiment can show reliability but not robustness. We removed "robustness" from the sentence.

31) • It may be worth commenting in the Summary & Discussion on the differences between the frame-dependent and the frame-independent diagnostic fields and how they should be used differently.

Lines 535-537 were expanded to read:

"It is interesting to note that the two frame-dependent methods – $L$ and $CD$, which were dominated by the large-scale gradient across the entire release domain and did not highlight any submesoscale features – were the least useful in identifying LCSs. For SPLASH drifters, this dominant large-scale gradient developed during the initial anticyclonic phase of motion, when drifters deployed closer/further from the center of rotation were shorter/longer and less/more complex. This overpowering trend could potentially be removed by moving into a co-rotating reference frame (i.e., a natural frame of reference), but identifying such natural reference frame is non-trivial in the absence of additional information about the flow."

32) References
• The references should be alphabetized.
• Filippi et al. (2021) should be given an 'a' and 'b' to differentiate the two publications.
• Froyland and Padberg-Gehle (2015) is not referenced in the manuscript.
• Essink et al. (2021) is missing from the references.
• It seems that the references for the supplemental materials are included here. If so, Beron-Vera et al. (2015) is also missing.

All comments implemented

33) Figures
• Across all figures (main and supplemental), the choice of colormaps is not ideal. The authors try to compare fields between different rows when they have different scales on the colormaps. For some quantities (e.g. FTLE), the colormap is strangely designed to highlight gradients in very specific narrow bands only. Some of the quantities (FTLE and dilation rate), the distinction of positive and negative values is important, but hard to do with the given colormaps. Thus, I recommend modifying the colormaps such that they vary continuously, except that for FTLE and dilation rate there is a clear break at 0.

We have improved the colormaps to show a more clear distinction between positive/negative values, and used standard Matlab's "parula" colormap (most common choice in scientific applications) instead of "jet" colormap to avoid highlighting narrow bands.

I also recommend using uniform ranges on the colorbars for quantities that are scaled by or independent of the time interval (FTLE, correlation dimension, dilation rate, LAVD).

It is sometimes difficult to see the details when uniform colorbars are used. There is a trade-off between using same vs. different ranges across different time-intervals, which allows for an easier comparison top to bottom vs. visualizing more details at any given time. In cases where enough details were visible with the same colorbar range, we now used the same range.

For the clusters, the number of different colors in the colormap should equal the number of clusters. It would also help if they were chosen to be more easily distinguishable, as currently some of the shades of blue and some of the shades of red are hard to tell apart.

We have adjusted the colorbars to have as many colors as there are clusters.

• Aside from providing the colorbars, the right column of the figures is not needed, ince the drifter positions are placed in geographic context in Figs. 1 and S2.

We decided to keep the right panels in the paper to avoid the need to go back to Fig. 1 every time. A zoomed-out view in the right panel sometimes helps visualizing the dominant patterns as well. However, in the Supplementary Material, we now replaced the right panels with an overlayed plot to simplify comparison between the SPLASH-like and dense-grid simulations in Figs. S3-S8. Since it is very hard to overlay the clusters, we kept right panels in Figs S9-S10 as before.

• Figs. 7 and 8: It would help to have a reminder in the caption here what the different symbols mean and why some diamonds are left white.

We have added this info to the caption: "Colored circles/colored diamonds/empty (white) diamonds mark drifters for which calculation of gradients was most reliable /less reliable/not reliable, according to the LLS reliability criteria described in the methods section)."

34) Supplementary materials
• Lines 21 – 24: Doesn't the topography also play a role here, in addition to the density gradients? And wind?

Yes, we rewrote the paragraph as follows:

"Density variations, topography, and wind forcing are generally all important factors in setting the circulation and transport in this geographical region of the Gulf of Mexico. Inspection of the model density fields (Fig. S1) suggested that 2 density features had a particularly strong influence on the movement of SPLASH drifters: the dense water filament (red) that affected the northern group of drifters during the splitting event around days 0.9-1.25, and the light water coastal plume (blue) that affected the southern group of drifters from day 1.25 and onward."

35)• Line 51: Why were different time intervals chosen in the model than for the observations? Line 47 suggests possibly using 1.5 days instead of 1 day, but why 2 days, and then 4 days?

We now see that our original wording might have been misleading and have clarified the text as follows:

"Just like the real drifters, the simulated drifters from the shifted release initially started moving anticyclonically around the recirculation feature towards the coast. The drifters then split into a smaller northern and larger southern group, the northern group aligned along the coast, and then the entire drifter ensemble proceeded offshore, all in qualitative agreement with real drifters. However, simulated drifters moved slower than real drifters during the initial and intermediate stages, i.e., it took them longer to approach the coast and form the along-shore filament. By day 1.5, simulated drifters started approaching the coast but they didn't fully form the characteristic filament until day 2. After that, their off-shore movement was comparable to that for real drifters. Thus, we chose 2 days for simulated drifters (instead of 1 day for real drifters) as the representative time for the intermediate stage, and then allowed them the same amount of time

(i.e., 2 days since formation of filament) to progress offshore as we did for real drifters, thus yielding 4 days as the analysis time for the third stage."

36) • Lines 63 – 64: I don't see the similarities in the patterns of high and low values. At early times: observations – scattering of a few high values near the corners of the deployment; model – coherent swath of high values in the center of the domain. At late times: observations – mostly high values throughout the domain, with a region of lower values in the lower half away from the edges; model – mostly low values, with a swath of very high values along the SE edge. The magnitude of the values is also not comparable, if one can use the ranges on the colorbars as a guide, especially for negative FTLE values.

We only meant that the distribution of drifters bore some resemblance, so we clarified that:

"The distribution of drifters at 0.5 days in the top middle panel of S3 bears some resemblance to that in Fig. 3 in that the separations between dots in the middle of the domain were larger than in the east and west. However, FTLEs were much smoother in the model and almost exclusively positive, compared to noisier mixed-sign FTLEs in observations."

37) • Line 83: It is not clear in what ways the agreement was favorable. E.g., one could not identify the coherent structures at the middle time from the SPLASH-like simulation alone, even if they are weakly reflected in it.

We clarified as follows:
"Overall, agreement between the SPLASH-like and dense-grid FTLEs was still favorable in the west, where SPLASH-like fields correctly indicated, or at least hinted upon, the high-FTLE ridge entering into the domain, and to the northwest of it, where FTLEs were uniformly lower, but not so good in the east/northeast."

38) • Line 96: I am not clear what standards of agreement are being used here. The structures in the dense simulation (S-shaped and longitudinal ridges) are not identifiable in the SPLASH-like simulation. Magnitudes are impossible to compare, since different colorbar ranges are used. Maybe a scatter plot of values at the positions of the SPLASH-like data from the two simulations would make that clearer. (Such scatter plots could replace the right columns of Figs. S3 – S8 to directly assess the reliability of the sparse calculations for all the Lagrangian diagnostics.)

As suggested by the reviewer, we have replaced the right panels with a new plot where we superimposed SPLASH-like FTLEs onto the dense-grid fields using the same color scheme. We like this option better than the scatter plot because it provides information about the

spatial distribution of the differences.  We have replaced the right panel with such an overlayed plot for other metrics and times in Figs S3-S8 as well.

The new overlayed plot makes it clear that:

"SPLASH-like fields at the 3ʳᵈ stage correctly indicated the presence of a manifold (region of largest FTLEs) near the northern half of the southeastern edge of the release domain. However, all other, narrower FTLE ridges visible in the dense-grid FTLEs were not resolved by the SPLASH-like release."

39) • Line 109: The SPLASH-like fields shows patches of increased $V_{en}$ similar to those deemed meaningful in the observations. Why are they considered negligible here?

We thank the reviewer for this comment and have implemented it into the text:

"At early times, SPLASH-like encounter volume fields showed patches that did not have any counterpart in the dense-grid fields, which were flat over most of the domain. This suggests that encounter volume might not be a good option for identifying coherent structures over short time intervals when drifters did not yet have enough time to encounter many neighbors, and that SPLASH-based encounter volume fields were severely affected by the limited number of drifter."

40) • Line 125: Are they confined to a smaller area, or are they hidden behind the large markers? There are some cyan dots hidden behind dark blue in the southernmost filament, but nearly all red values are confined to near 90W and 28.82N.

41) • Line 156: I would not say that the SPLASH-like simulation successfully mapped out the dominant features at 4 days, although it did seem to show a hint of the region of lower LAVD values in the middle.

We agree. The new text now reads:

"The main features of the dense-grid *LAVD*s at the intermediate and later stages were the blue cluster near the middle of the domain at 89.53W, 29.3N, the yellow filament entering the domain

from the western corner, a small yellow cluster in the very south, and another yellow cluster near

the southeastern edge of the domain (which coincided with the area to the east of the letter-S-

shaped filament identified in the FTLE maps). None of the yellow clusters were resolved in

SPLASH-like maps, and only a hint of the low-$LAVD$ region in the middle of the domain was

suggested by the SPLASH-like fields. Overall, SPLASH-like $LAVD$ maps were reliable at early

times, as suggested by the good agreement with their dense-grid counterparts, but deteriorated

significantly at later times, to the point of missing all features except a low-$LAVD$ region in the

middle of the domain."

42) • Line 168: Fig. S9 shows only 4 clusters for the second version at intermediate
times. Yes, we corrected this typo, thank you.

43) • Line 191: Is it meaningful to compare the number of clusters found in the two
simulations over different domains? Maybe it would be better to apply the spectral
clustering in the dense simulation only to the subdomain also sampled by the
SPLASH-like simulation.

The domain used in the dense-grid SC simulations has same minimum and maximum
values for latitude and longitude as SPLASH domain, but is rectangular rather than
rhomboid. As some of the LCS, such as the S-shaped manifold in the east and longitudinal
manifold in the south, were confined to near the edges of the release domain, we decided to
keep these regions in the dense-grid simulations.

44) • Line 203: I disagree that the clusters look similar in the figures. The only similarity
I find is the split around 29N.

We rewrote the paragraph, taking into account reviewer's comment:

"To summarize, at early times, both SPLASH-like and dense-grid simulations resulted in

splitting the SPLASH release domain into a large number of clusters, although the exact splitting

was different between the two simulations. At later times (both at intermediate and late stages),

the number of clusters present within the SPLASH domain dropped significantly, although

dense-grid simulations also identified several additional clusters in the south and in the

northwestern corner, all laying outside of the SPLASH domain. At the intermediate stage, both

SPLASH-like and dense-grid simulations produced a configuration with the split along 29N. And at later stage, both SPLASH-like and dense-grid simulations produced a configuration where most of the SPLASH domain was assigned to just one cluster (blue)."

45) • Line 213: I suggest citing the filtering method developed for these kinds of trajectories by Yaremchuk & Coelho (2015, doi: 10.1109/JOE.2014.2353472).

We thank the reviewer for pointing out a relevant reference, which we now cite in the revision.

46) • To draw comparisons with the observations (e.g. "similar magnitudes"), it would be helpful to use the same colormaps and ranges in the supplemental figures as are used in the main manuscript for the corresponding diagnostics.

In the revision, we do so whenever possible, i.e., in all simulations where enough details are visible in the figures when the same colormap is used as in observations.

47) • Fig. S4: I don't understand why the values shown along the longitudinal filament in the south at the late time don't agree between the two simulations. Coarse sampling should have no impact on pathlength. Is this a plotting artifact?

Yes, this is a plotting artefact, as the red dots in the filament in bottom panel are hidden behind the blue dots. We now explain this in the text. The new right subplot, where we overlay the SPLASH-like and dense-grid maps also helps to clarify this.

48) *Language*
Below are a few suggestions to fix grammar and spelling issues. Overall, this is one of the most readable manuscripts I have reviewed lately; so these are just minor things.

We thank the reviewer for the positive feedback on the readability of our paper. We have implemented all of the language improvement suggestions.

• Line 68: "and then use observations of their trajectories"
• Line 91: "parallels were drawn between…"
• Line 123: "coordinate system
• Line 150: "frame-independent"
• Line 210: "distance between the $i_{th}$ and $j_{th}$ trajectories"
• Line 216: "the cluster sizes"
• Line 225: "The LLS methods"
• Lines 226 and 229: The vectors for U and A should be transposed. (They need to be n x 1 and 3 x 1, respectively.)
• Line 240, 247: "LLS"
• Line 262: "Some clustering temporarily occurs" (no "of")

- Lines 338 – 339: "neither" and "nor" should be replaced with "either" and "or"
- Line 463: Strike "the" from "the NOAA's Global Drifter Program"
- Line 474: Strike "respectively" (respective to what?)
- Line 526: "The SPLASH experiment was…"
- Line 542: "In the SPLASH experiment"
- Line 710: "experiment sight" –site (not sight or cite)
- Line 731: "(bottom) 3 days"
- Supp, Line 8: "the site of the SPLASH experiment"
- Supp, Line 29: "as is evident"
- Supp, Line 91: "Fig. S3"
- Supp, Line 117: "blue values"
- Supp, Line 121: "a bit of red"
- Supp, Line 206: I suggest spelling out STD.
- Supp, Line 208: "$V_{en}$" instead of "$N_{en}$"

---

## Author Response (AR2)

**Editor's Comments to the author**:
Dear authors,

Congratulations, your paper has been accepted for publication in NPG. Don't forget to make the technical corrections highlighted by Review 1. Tanks again and I hope you will consider NPG for your next submissions.

Sincerely,
Pierre Tandeo

**Dear Editor, we have made the technical correction highlighted by Reviewer 1 (i.e., fixed the years of the two citations in line 285). Thank you. ~Irina**

**Reviewer's Report #1:** The authors thoroughly addressed all my previous concerns, and I recommend the revised manuscript for publication.

One remaining technical comment: On line 285, the publication years of the two citations appear to be switched.

**We thank the reviewer for finding this typo. We have fixed the years of the two citations in line 285.**